# Plants monitor the integrity of their barrier by sensing gas diffusion

Hiroyuki Iida[1✉], Isidro Abreu[2,5], Jennifer López Ortiz[1,5], Lucas León Peralta Ogorek[1,4,5], Vinay Shukla[2,5], Meeri Mäkelä[1], Munan Lyu[1], Alexey Shapiguzov[1,3], Francesco Licausi[2] & Ari Pekka Mähönen[1✉]

Barrier tissues isolate organisms from their surrounding environment. Maintaining the integrity of the tissues is essential for this function. In many seed plants, periderm forms as the outer barrier during secondary growth to prevent water loss and pathogen infection[1]. The periderm is regenerated when its integrity is lost following injury; however, the underlying mechanism remains largely unknown, despite its importance for plant survival. Here we report that periderm integrity in *Arabidopsis* roots is sensed by diffusion of the gases ethylene and oxygen. Following injury of the periderm, ethylene leaks out through the wound and oxygen enters, resulting in attenuation of ethylene signalling and hypoxia signalling. This condition promotes periderm regeneration in the root. When regeneration is complete and barrier integrity is re-established, pre-injury levels of ethylene and hypoxia signalling are regained. Gas diffusion monitoring is also used to re-establish the barrier in inflorescence stems after the epidermis is injured. We thus propose that gas diffusion is used by plants as a general principle to monitor and re-establish barrier integrity.

The periderm is a protective outer tissue established during secondary growth in many seed plants. It consists of several layers containing three different cell types: the phellem, phellogen and phelloderm[1,2] (Fig. 1a). The outermost cell type, phellem (also known as cork), differentiates from the outer daughter cells of dividing phellogen cells. The differentiating phellem cells deposit lignin and suberin in their cell walls to form a physical barrier to protect secondary tissues from biotic and abiotic stresses[1]. As the periderm is the interface between the plant's internal tissues and its environment, it is prone to injury. Periderm is re-established at wound sites in tree trunks, potato tubers and some fruits (such as apple)[3–5]. Periderm regeneration at wound sites is critical to prevent water loss and pathogen entry through wounds[4]. In addition to being essential for survival, the phellem of some tree species (such as cork oak) has been used by humans as manufacturing material for thousands of years. Understanding the mechanisms of periderm re-establishment would therefore have substantial biological and economic value. Even though the formation of periderm at wounds has been studied for more than a century, its underlying mechanism is still largely unknown.

The accumulation of the gaseous hormone ethylene controls plant growth and development. When ethylene diffusion is limited by the surrounding environment, the increased concentration serves as a developmental signal for plant tissues. For instance, ethylene diffusion is sensed by root tips to monitor soil compaction; in compact soil, ethylene accumulates in and near the root, resulting in fortified growth[6]. Aerenchyma formation and internode elongation in rice are induced by the accumulation of ethylene caused by limited diffusion under waterlogged conditions[7,8]. Here we show that the *Arabidopsis* barrier tissue, the periderm, limits the diffusion of two gaseous molecules, ethylene and oxygen, and we propose that monitoring the accumulation or depletion of these two gases functions as a system for maintaining periderm integrity.

## *Arabidopsis* roots regenerate periderm

To examine whether the periderm regenerates in *Arabidopsis* roots, we longitudinally cut the mature part of roots (Fig. 1b) and observed the morphological changes and the expression of the periderm reporter genes *PEROXIDASE15* (*PER15*), *PER49*, *PYK10-binding protein 1* (*PBP1*), *AT3G26450*, *WUSCHEL RELATED HOMEOBOX 4* (*WOX4*) and *AT1G14120* (refs. 9–12). We found that the reporters showing their expression in phellem (*PER15*, *PER49*, *PBP1* and *AT1G14120*) were induced at the wound site 1 day after injury (dai; Fig. 1c and Extended Data Fig. 1a,c). Phellogen characteristics appeared at 2 dai, as indicated by the appearance of the phellogen-preferred reporter *AT3G26450* and periclinal (that is, parallel to the cut surface) cell divisions at the wound site (Fig. 1c,d and Extended Data Fig. 1c). *WOX4* expression was detected less frequently at the wound site at 2 dai (4 out of 29 sections) and more consistently at 3 dai (Fig. 1c and Extended Data Fig. 1c). At 4 dai, the surface-exposed cells showed lignification and suberization, indicating that phellem-like layers were established at the wound site (Fig. 1d and Extended Data Fig. 1b). Phellem-like cells were adjacent to the inner cells that were actively dividing, thus resembling phellogen. These observations show that periderm regenerates at the wound site in *Arabidopsis* roots.

Next we examined the functionality of the re-established barrier. We used *proPXY:GUS*, in which the *GUS* (β-glucuronidase) gene is expressed

[1]Organismal and Evolutionary Biology Research Programme, Faculty of Biological and Environmental Sciences, Viikki Plant Science Centre, University of Helsinki, Helsinki, Finland. [2]Department of Biology, University of Oxford, Oxford, UK. [3]Natural Resources Institute Finland (Luke), Production Systems, Piikkiö, Finland. [4]Present address: School of Biosciences, University of Nottingham, Loughborough, UK. [5]These authors contributed equally: Isidro Abreu, Jennifer López Ortiz, Lucas León Peralta Ogorek, Vinay Shukla. ✉e-mail: hiroyuki.iida@helsinki.fi; aripekka.mahonen@helsinki.fi

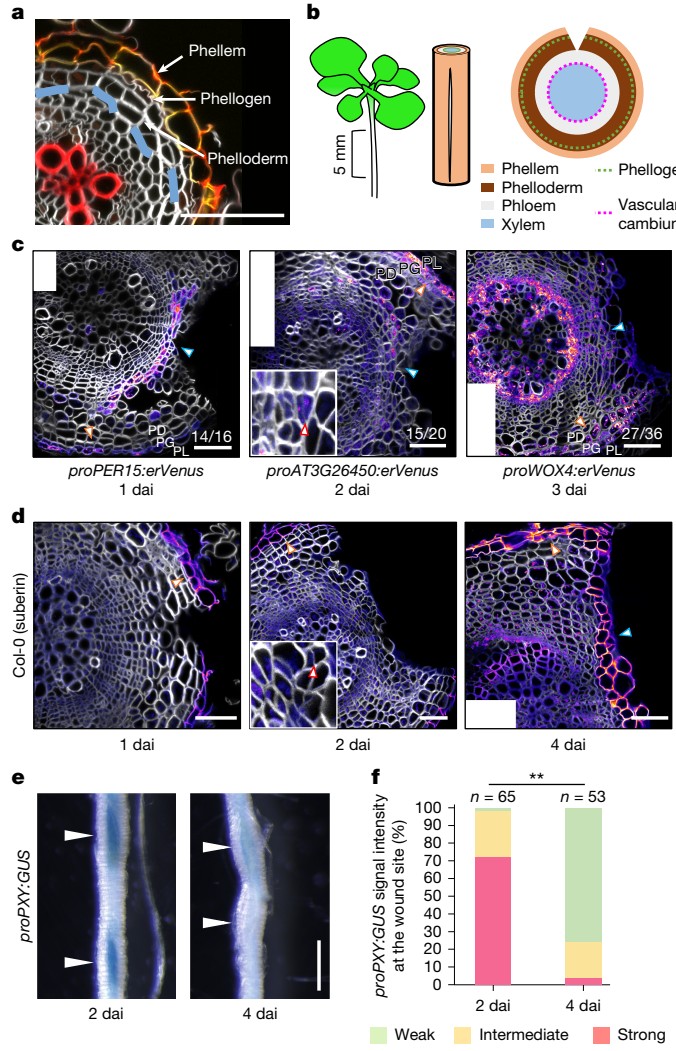

**Fig. 1 | Mechanical injury elicits periderm regeneration from vascular tissues in *Arabidopsis* roots. a**, A cross-section of wild-type roots. Lignin (red) and suberin (yellow) were stained. **b**, Schematic of the wounding experiment. Roots within 5 mm below the root–hypocotyl junction were cut longitudinally (light brown, phellem; green dotted line, phellogen; dark brown, phelloderm; grey, phloem region; magenta dotted line, vascular cambium; blue, xylem region). **c**, The promoter activity of *PER15* (left), *AT3G26450* (centre) and *WOX4* (right) at 1, 2 and 3 dai, respectively. In the intact periderm, the promoter activity of *PER15*, *AT3G26450* and *WOX4* was preferentially detected, respectively, in both the phellogen (PG) and young phellem (PL), in the dividing cells (presumably the phellogen), and in the phellogen and strongly in the phelloderm (PD). **d**, Cross-sections of wild-type roots at 1, 2 and 4 dai of 17-day-old roots. Col-0, Columbia-0. In **c**,**d**, orange and blue arrowheads indicate the normal periderm or the wound sites, respectively; red arrowheads point to thin cell walls parallel to the wound surface, indicating recent cell divisions at 2 dai; the insets are the magnified images of the wound sites. White rectangles mark empty corners of stitched images. **e**, *proPXY:GUS* signals at the wound site at 2 dai and 4 dai of 17-day-old roots. White arrowheads indicate the wound sites. **f**, *proPXY:GUS* signal strength at the wound site in the seedlings at 2 and 4 dai. *n* indicates the number of examined wound sites. Two-sided Fisher's exact test was used (**P < 0.01). Venus signal intensities in **c** and intensity of suberin staining with Fluorol yellow in **d** are shown according to the colour scales on the right. The top (brighter) area of the scale represents a higher intensity of signals. White, SR2200 (cell wall). Fractions on the panels indicate the proportion of cross-sections showing a similar expression as in the images. Scale bars, 50 μm (**a**,**c**,**d**) and 0.5 mm (**e**).

under the control of the *PHLOEM INTERCALATED WITH XYLEM* (*PXY*) promoter, which is active in the vascular cambium and xylem parenchyma, the inner tissues of the mature root[13] (Extended Data Fig. 1d). Collected roots were incubated in a buffer containing the GUS substrate X-Gluc. We reasoned that an intact periderm would inhibit the entry of X-Gluc into the mature root, resulting in reduced GUS signal. Consistent with this, the GUS signal was stronger in the wounded region than in the intact region of *proPXY:GUS* roots at 2 dai (Fig. 1e). GUS signal intensity became weaker at 4 dai, indicating that the integrity of the re-established periderm barrier was sufficiently restored to prevent X-Gluc penetration (Fig. 1e,f). The change in GUS signal levels was not caused by changes in promoter activity, as *proPXY:erVenus* fluorescence levels remained unchanged after the injury and during regeneration (Extended Data Fig. 1d,e). Taken together, our results show that phellem identity, as indicated by the expression of phellem markers, is established at 1 dai, followed by phellogen-like cell divisions at 2 dai and phellem differentiation (deposition of lignin and suberin) by 4 dai, coinciding with regained barrier function.

## Ethylene diffusion promotes regeneration

Next we examined the mechanisms underlying periderm regeneration. As plants sense injury as stress[14], we investigated whether stress-related hormones affect the induction of *PER15*, one of the periderm genes induced early in regeneration. *proPER15:erVenus* seedlings were treated with methyl jasmonate, abscisic acid or 1-aminocyclopropane-1-carboxylate (ACC), a precursor of ethylene, for 1 dai. Whereas neither methyl jasmonate nor abscisic acid treatment affected *PER15* induction, ACC treatment significantly reduced induction (Fig. 2a,b and Extended Data Fig. 1g,h). Using other reporter lines, we found that the induction of most periderm markers was also reduced by ACC treatment at 1 dai (Extended Data Fig. 2a,b,e). We also found that *PER15* and *PBP1* expression and phellem formation were not affected in intact roots following ACC treatment (Extended Data Fig. 2c,d,f,g). Therefore, it seems that there are other signals for periderm development in intact tissues. Although ACC could act independently from ethylene signalling[15,16], ACC treatment did not affect *PER15* induction at 1 dai in *ethylene insensitive 2* (*ein2-1*) and *ethylene response 1* (*etr1-3*) mutants. We also showed that ethylene treatment reduced *PER15* and *PBP1* induction at the wound site (Extended Data Fig. 2l,m,o). Taken together, these results indicate that canonical ethylene signalling is required to suppress periderm gene induction (Extended Data Fig. 2h–m,o). We next assessed whether ACC treatment affects suberized cell formation at the wound site. Whereas suberized cells in control roots formed a continuous layer at the wound site, treatment with ACC after injury occasionally resulted in discontinuous suberized cell layers or callus-like structures at the wound site (Fig. 2c,d). The discontinuous suberized cell layers at the wound site were also found following ethylene treatment (Extended Data Fig. 2n,o). To examine the functionality of the re-established barriers in ACC-treated roots, we tested X-Gluc penetration. Even though *PXY* promoter activity was repressed following ACC treatment at 4 dai, *proPXY:GUS* signals near the wound site were stronger in ACC-treated roots than in the untreated control at 4 dai (Extended Data Fig. 1d–f). The combination of stronger GUS signals and reduced promoter activity indicates that the barrier is less functional in ACC-treated roots. Altogether, these results show that ethylene impedes periderm re-establishment at the wound site.

As it has been reported that ethylene production is increased after mechanical injury[17], the suppression of periderm regeneration by ethylene seems counter-intuitive. To examine the ethylene signalling level, we generated a dynamic ethylene reporter line, *RPS5A:erVenus-EBF1UTR*, in which *erVenus* fused with the 3′ untranslated region (UTR) of *EIN3-BINDING F BOX PROTEIN 1* (*EBF1*) is expressed under the control of the constitutive *RPS5A* promoter (Extended Data Fig. 3a,e). As *erVenus-EBF1UTR* translation is inhibited by the EIN2 protein in the

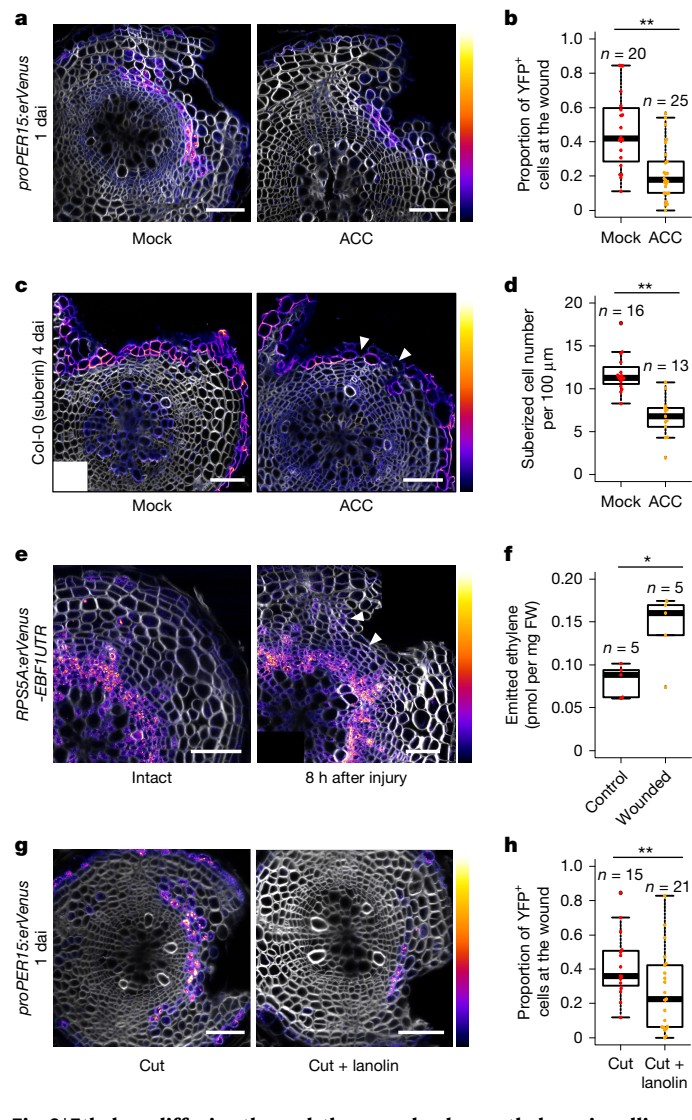

**a** *proPER15:erVenus* 1 dai

Mock | ACC

**b**

Proportion of YFP+ cells at the wound

$n = 20$ | $n = 25$

**

Mock | ACC

**c** Col-0 (suberin) 4 dai

Mock | ACC

**d**

Suberized cell number per 100 μm

$n = 16$ | $n = 13$

**

Mock | ACC

**e** *RPS5A:erVenus -EBF1UTR*

Intact | 8 h after injury

**f**

Emitted ethylene (pmol per mg FW)

$n = 5$ | $n = 5$

*

Control | Wounded

**g** *proPER15:erVenus* 1 dai

Cut | Cut + lanolin

**h**

Proportion of YFP+ cells at the wound

$n = 15$ | $n = 21$

**

Cut | Cut + lanolin

**Fig. 2 | Ethylene diffusion through the wound reduces ethylene signalling and triggers periderm regeneration. a**, Cross-sections of 18-day-old *proPER15:erVenus* roots at 1 dai grown on MS plates supplemented without (mock) or with 10 μM ACC (ACC) for 1 dai. **b**, The proportion of cells at the wound site showing Venus yellow fluorescent protein (YFP) signal intensities above the threshold was quantified at 1 dai in mock- or ACC-treated 18-day-old *proPER15:erVenus* roots. **c**, Cross-sections of 21-day-old wild-type roots at 4 dai grown on MS (mock) or 10 μM ACC-supplemented MS (ACC) plates for 4 dai. White box marks empty corner of stitched image. White arrowheads indicate gaps in the suberized cell layer. **d**, The density of suberized cells at the wound site was quantified in mock- and ACC-treated 21-day-old wild-type seedlings at 4 dai. **e**, Cross-sections of 17-day-old *RPS5A*:*erVenus-EBF1UTR* roots without injury or 8 h after the injury. White arrowheads indicate an increase in Venus signals. **f**, Concentration of ethylene emitted from roots 3 h after injury. The mature part of the wild-type roots was intact (control) or injured (wounded). FW, fresh weight. **g**, Cross-sections of 18-day-old *proPER15:erVenus* roots at 1 dai grown without (cut) or with lanolin (cut + lanolin) at the wound for 1 dai. **h**, The proportion of cells at the wound site showing Venus signal intensities above the threshold was quantified in 18-day-old *proPER15:erVenus* seedlings at 1 dai. Two-tailed Wilcoxon rank-sum test was used in **b**,**d**,**h**, and two-tailed Welch's *t*-test was used in **f** (*$P < 0.05$, **$P < 0.01$). *n* indicates the number of examined cross-sections in **b**,**d**,**h** and the number of repeats in **f**. For descriptions of the different elements for all box plots, see Methods, 'Statistics and reproducibility'. Venus signal intensities in **a**,**e**,**g** and intensity of suberin staining with Fluorol yellow in **c** are shown according to the colour scales on the right. The top (brighter) area of the scale represents a higher intensity of signals. White, SR2200 (cell wall). Scale bars, 50 μm.

presence of ethylene[18], the Venus signal will be detected only in cells in which the ethylene signalling level is low. Thus, low Venus fluorescence levels indicate high ethylene signalling levels. We validated this reporter line in both the root apical meristem and intact or wounded root secondary tissues using ACC and inhibitors of ethylene biosynthesis or signalling, aminoethoxyvinylglycine (AVG) and AgNO3, respectively. Validation confirmed that *RPS5A:erVenus-EBF1UTR* accurately reports ethylene signalling levels (Extended Data Figs. 3b,c,f–i and 4a). Following periderm injury, *RPS5A:erVenus-EBF1UTR* signal intensity was greater (that is, ethylene signalling lower) in the phloem parenchyma near the wound at 2 dai (Extended Data Fig. 3d,k). In the control line *RPS5A:erVenus*, expression was not affected by injury, indicating that the increase in the *RPS5A:erVenus-EBF1UTR* signal was because of enhanced *erVenus-EBF1UTR* translation rather than enhanced *RPS5A* promoter activity (Extended Data Fig. 3d,j). These observations demonstrate that ethylene signalling is reduced after wounding, which is consistent with the negative role of ethylene in periderm regeneration.

Altogether, our data indicate the existence of a mechanism to reduce ethylene signalling following wounding. This led us to reason that ethylene gas is released from the wound into the surrounding environment. Lignin and suberin in differentiated phellem normally prevent gas exchange through the surface, allowing ethylene to accumulate in unwounded secondary tissues. Following periderm injury, ethylene leaks through the wound, decreasing ethylene signalling and thus triggering periderm regeneration. To test this hypothesis, we examined expression changes in the ethylene signalling reporter line at earlier time points, as ethylene diffusion should occur right after injury. *RPS5A:erVenus-EBF1UTR* signal intensities increased in the exposed distal phloem region already at 8 h and more clearly at 11 h after injury, demonstrating a rapid reduction in ethylene signalling levels after injury (Fig. 2e and Extended Data Fig. 4b). We also measured ethylene emission by roots using gas chromatography. We detected higher ethylene emission from wounded roots compared with control roots at all measured time points (Extended Data Fig. 4c). The ethylene concentration was significantly higher 3 h after injury (Fig. 2f), indicating an increase in emitted ethylene immediately after injury. We also investigated reporter gene expression and periderm re-establishment when gas leakage was prevented by covering the wound with lanolin or Vaseline or by submerging the seedlings in liquid Murashige and Skoog (MS) medium. The *RPS5A:erVenus-EBF1UTR* signal was weaker at the wound site in covered roots than in roots with uncovered wounds at 1 dai (Extended Data Fig. 5b). The reduction was not due to reduced promoter activity, as *RPS5A:erVenus* remained unaltered after covering the wound site (Extended Data Fig. 5a). These observations demonstrate that a high level of ethylene signalling is maintained when a wound is physically sealed. We consistently found that most periderm marker induction at 1 dai and suberized cell layer formation at 4 dai failed to occur when the wound was sealed with lanolin immediately after wounding (Fig. 2g,h and Extended Data Fig. 5c–e). *PER15* activation was also decreased at 1 dai when seedlings were submerged in liquid MS immediately after injury to limit gas exchange (Extended Data Fig. 5f). Altogether, these results indicate that ethylene diffusion from the wound leads to a decrease in its signalling level, and this reduction promotes periderm regeneration.

## Oxygen flows into tissues after injury

As ACC treatment did not fully inhibit suberized cell formation at the wound site, we inferred that there might be other, probably gaseous, regulators involved in periderm regeneration. We considered oxygen as a candidate. Oxygen is consumed in cellular respiration during tissue growth, but external oxygen cannot easily enter secondary tissues because of the poor permeability of phellem layers. We therefore reasoned that secondary tissues are normally under physiological hypoxic conditions and that oxygen flows into the tissue

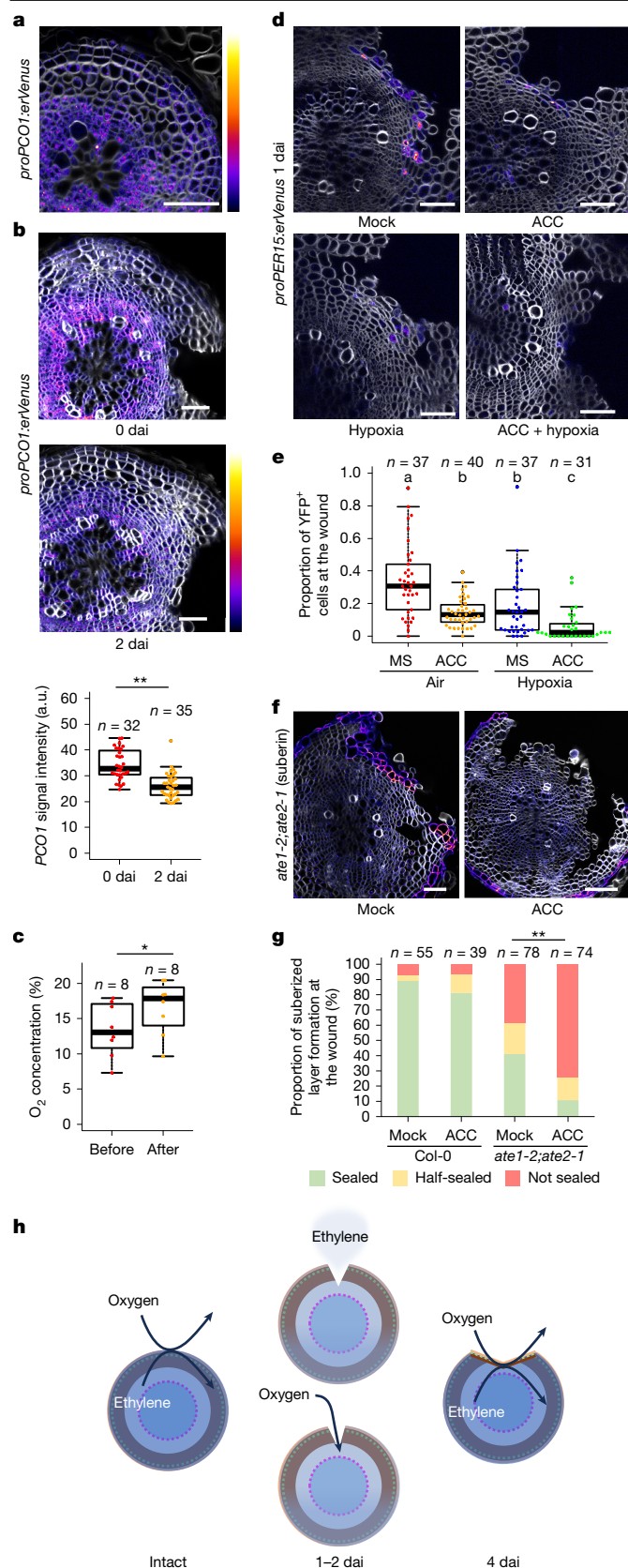

**Fig. 3 | Oxygen entry through the wound suppresses hypoxia signalling and promotes periderm regeneration. a**,**b**, Cross-sections of 14-day-old intact *proPCO1:erVenus* roots (**a**) and 21-day-old *proPCO1:erVenus* roots right after injury (**b**, top) or at 2 dai (**b**, middle); and quantification of *proPCO1:erVenus* signals in the vascular region at 0 and 2 dai (**b**, bottom). Two-tailed Welch's *t*-test was used (**$P < 0.01$). **c**, The quasi-steady-state oxygen concentration was measured and quantified before and after the periderm peeling. Paired Wilcoxon rank-sum test was used (*$P < 0.05$). **d**, Cross-sections of 25-day-old *proPER15:erVenus* roots at 1 dai grown on MS (mock) or 10 µM ACC-supplemented MS (ACC) plates in ambient air, or on MS (hypoxia) or 10 µM ACC-supplemented MS (ACC + hypoxia) plates under 5% oxygen concentration for 1 dai. **e**, The proportion of cells at the wound site showing Venus signal intensities above the threshold was quantified at 1 dai in mock ACC-treated, hypoxia-treated and ACC + hypoxia-treated *proPER15:erVenus* roots. Kruskal–Wallis test followed by Dwass–Steel–Critchlow–Fligner pairwise comparisons was used (different letters indicate statistically significant differences between two groups; $P < 0.01$). **f**, Cross-sections of 21-day-old *ate1-2;ate2-1* roots at 4 dai grown on MS (mock) or 10 µM ACC-supplemented MS (ACC) plates for 4 dai. **g**, The proportion of suberized cell formation at the wound site in mock and ACC-treated 21-day-old wild-type and *ate1-2;ate2-1* roots at 4 dai. Two-sided Fisher's exact test was used to test for significant differences between mock- and ACC-treated *ate1-2;ate2-1* (**$P < 0.01$). **h**, Schematic of periderm integrity surveillance mechanisms mediated by ethylene and oxygen diffusion. *n* indicates the number of examined cross-sections in **b**,**e**,**g** and the number of repeats in **c**. Venus signal intensities in **a**,**b**,**d** and intensity of suberin staining with Fluorol yellow in **f** are shown according to the colour scales on the right. The top (brighter) area of the scale represents a higher intensity of signals. White, SR2200 (cell wall). Scale bars, 50 µm.

conditions[19]. The *PCO1* and *PCO2* promoters were active in mature roots, and reporter expression was detected more strongly in the vascular region and the periderm, and weakly in the distal phloem parenchyma (Fig. 3a and Extended Data Fig. 6a). This result supports the hypothesis that inner cells in the mature root experience hypoxic conditions. Next we examined hypoxia signalling in the root after injury. The activity of both *PCO1* and *PCO2* promoters was weaker in whole tissues at 2 dai compared with immediately after injury (Fig. 3b and Extended Data Fig. 6b). We detected a reduction in *PCO1* and *PCO2* promoter activities already 16 h after the injury (Extended Data Fig. 6c), suggesting that oxygen entry quickly alleviates the hypoxic conditions. To examine whether the periderm affects the entry of oxygen, we measured the steady-state oxygen level by inserting an oxygen microsensor[20] into wild-type secondary tissue (Fig. 3c). After the measurement, we partially peeled off the periderm and inserted the microsensor into the peeled region. Comparing the measurements before and after the removal of the periderm, we found that the oxygen level was higher in the root after the removal of the periderm, indicating that the periderm prevents the entry of oxygen (Fig. 3c and Extended Data Fig. 6d,e). Altogether, these data support the idea that oxygen enters through the wound and inhibits the hypoxic response in secondary tissues.

## Ethylene and hypoxia act additively

To test whether a reduction in hypoxia signalling is required for periderm regeneration, we grew *proPER15:erVenus* seedlings under 5% oxygen concentration for 1 dai. We found that hypoxia treatment significantly reduced *PER15* induction at the wound site, suggesting that *PER15* induction requires a decrease in hypoxia signalling (Fig. 3d,e). To further validate the role of hypoxia signalling, we used *arginine transferase* (*ate1;ate2*) and *proteolysis6* (*prt6*) mutants that show a constitutively active hypoxia response because the target proteins for the N-degron pathway, which includes hypoxia signalling regulators, are stable in these mutants regardless of the oxygen level[21–24]. At 4 dai, *ate1-2;ate2-1* roots often showed only half or no suberized layer formation at the wound site (Fig. 3f,g). These findings imply that maintaining a high level of hypoxia signalling inhibits periderm regeneration.

through the wound following periderm injury and promotes periderm regeneration.

To investigate this hypothesis, we generated transcriptional reporter lines of *PLANT CYSTEINE OXIDASE* (*PCO*) genes (*proPCO1:erVenus* and *proPCO2:erVenus*) whose expression is increased under hypoxic

To examine whether ethylene and hypoxia signalling additively regulate periderm regeneration, we investigated *PER15* induction following ACC treatment under hypoxia. As ACC oxidase requires oxygen to convert ACC into ethylene, we first studied whether ACC can increase ethylene signalling in the hypoxia conditions we used. ACC treatment under ambient conditions reduces primary root growth[25], promotes root hair formation[26] and stabilizes *35S:EIN3-GFP* in the root tips[27]. We found that a 1-day treatment with ACC separately or combined with hypoxia showed similar growth defects and EIN3–GFP induction (Extended Data Fig. 6h), indicating that ACC treatment activates ethylene signalling even in our low-oxygen conditions. When plants were treated with ACC for 1 dai under hypoxic conditions, *PER15* induction at the wound site was more severely suppressed than with ACC treatment or hypoxia alone (Fig. 3d,e). To further examine the combined effect, we treated *prt6-5* and *ate1-2;ate2-1* mutants with ACC after injury. Whereas untreated *prt6-5* roots had suberized cell formation similar to the wild type at the wound site, ACC-treated *prt6-5* roots showed more frequent formation of callus-like structures (Extended Data Fig. 6f,g). In addition, most ACC-treated *ate1-2;ate2-1* mutant roots did not form suberized cells at the wound site (Fig. 3f,g). These results demonstrate that the reductions in ethylene and hypoxia signalling act additively in periderm regeneration (Fig. 3h).

We also examined whether the normal (that is, not regenerated) periderm formation is affected in *ate1-2;ate2-1* mutants treated with ACC during secondary development. We did not find defects in suberization in the phellem, indicating that normal periderm formation is more robust than regeneration and is regulated by factors other than ethylene and oxygen (Extended Data Fig. 2g).

The additive effect between ethylene and hypoxia is consistent with the strong suppression of *PER15* induction at the wound site when seedlings were submerged, which results in high levels of ethylene and hypoxia signalling after injury (Extended Data Fig. 5f). To examine whether lowering either ethylene or hypoxia signalling is sufficient to induce *PER15* expression at the wound site of submerged seedlings, we used e*in2-1* or *etr1-3* and fully oxygenated liquid MS medium. Whereas *proPER15:erVenus* showed less frequent *PER15* induction at the wound site in wild-type roots submerged in control liquid MS at 1 dai, we detected clear but variable *PER15* induction following injury when the plants were submerged in oxygenated liquid MS, suggesting that supplementing oxygen partially rescues *PER15* induction (Extended Data Fig. 7a,d). In the *ein2-1* background, oxygenated liquid MS medium more stably rescued *PER15* induction at the wound site compared with the case in the wild type (Extended Data Fig. 7b,e). We also observed a slight but not significant rescue in the *etr1-3* background. (Extended Data Fig. 7c,f). Furthermore, we examined suberization of cells under submerged conditions. Liquid MS medium without gas supplementation reduced the level of suberized cell formation at the wound site in wild-type, *ein2-1* and *etr1-3*. However, we found that when submerged in aerated or oxygenated liquid MS medium, the ethylene signalling mutant roots showed a significant increase in the density of suberized cells at the wound site compared to wild-type roots grown under liquid MS medium without gas supplementation (Extended Data Fig. 7g–j). Overall, these results indicate that lowering ethylene or hypoxia signalling rescues periderm regeneration under submerged conditions that prevent the rapid diffusion of gases.

Next we investigated the mechanism that terminates periderm regeneration. When the periderm is re-established at the wound site, gas diffusion once again becomes limited, which may be the cue to terminate regeneration. To determine whether diffusion of ethylene and oxygen is suppressed after periderm re-establishment, we observed the expression of ethylene and hypoxia signalling reporters. *RPS5A:erVenus-EBF1UTR* signal intensity decreased (that is, ethylene signalling increased) in the phloem parenchyma near the wound at 5 dai compared with 2 dai, although the *RPS5A* promoter activity remained unchanged (Extended Data Fig. 8a,b,e). The *proPCO1:erVenus* and *proPCO2:erVenus* signals were also stronger in whole tissues at 4 dai than at 2 dai (Extended Data Fig. 8c,d,f). Thus, both hypoxia and ethylene signalling returned to pre-injury levels, supporting the idea that diffusion of ethylene and oxygen is reduced after the periderm is re-established (Fig. 3h).

We next examined the phenotypes of *ein2-1* and *etr1-3* mutants, in which ethylene signalling is inactive even after periderm re-establishment, to determine whether this results in a failure to terminate periderm regeneration. At 6 dai, *ein2-1* roots showed a significant increase in the density of suberized cells at the wound site (Extended Data Fig. 9a,b), suggesting a failure to precisely terminate the periderm regeneration. To study the possible role of wound-induced ethylene biosynthesis in regeneration, we used ethylene biosynthesis (AVG) and signalling inhibitors (AgNO$_3$). These inhibitors had no effect on *PER15* and *PBP1* induction at 1 dai, with the exception of a slight reduction in *PER15* expression following AVG treatment. These results indicate that ethylene biosynthesis following wounding may not be required for periderm gene activation and that the further reduction in ethylene signalling level does not enhance the induction of periderm genes (Extended Data Fig. 9c,d,f,g). However, consistent with the phenotype in *ein2-1* roots, our findings showed an increased density of suberized cells at the wound site at 6 dai following the inhibitor treatment (Extended Data Fig. 9e,h). These findings support the hypothesis that ethylene signalling is required to precisely terminate the phellem differentiation process.

## Gas-mediated barrier monitoring in shoot

The above results indicate that root periderm integrity is monitored through diffusion of ethylene and oxygen. We therefore examined whether the integrity of other barriers is monitored in the same way. *Arabidopsis* inflorescence stems do not develop periderm or suberized cell layers; instead, the epidermis with cuticle layer acts as a barrier[28]. When inflorescence stems were longitudinally cut (Fig. 4a), we found that expression of *PER15* and *PER49* (but not other examined periderm genes) appeared at the wound site 1 dai, and a suberized cell layer was established at the wound site 4 dai (Fig. 4b and Extended Data Fig. 10a). These observations indicate that a phellem-like layer is formed at the wound site in the inflorescence stems. Next we tested whether the re-establishment of a barrier in inflorescence stems is also mediated by the diffusion of gases. In accordance with this idea, sealing the wound with lanolin or Vaseline inhibited *PER15* induction at 1 dai and suberized cell formation at 4 dai (Fig. 4c–e).

To examine whether ethylene and/or oxygen diffuse through the wound to promote barrier re-establishment in inflorescence stems, we first investigated hypoxia signalling. Unlike in roots, *PCO1* and *PCO2* expression was maintained, and *ate1-2;ate2-1* stems did not show defects in suberized cell layer formation following injury (Extended Data Fig. 10b,c). There was thus no clear indication of either reoxygenation following injury or a role for hypoxia signalling in barrier re-establishment in inflorescence stems. We next examined ethylene diffusion. We found that, as in roots, wounded stems emitted a higher concentration of ethylene than unwounded controls (Fig. 4f and Extended Data Fig. 10d). Furthermore, when wounds were sealed with Vaseline immediately after injury, we detected variable but weaker *RPS5A:erVenus-EBF1UTR* signals in the stems at 2 dai compared with controls (Fig. 4g,h and Extended Data Fig. 10e,h,i). These results indicate that gaseous molecules, such as ethylene, may diffuse through the wound. We therefore investigated the role of ethylene in barrier re-establishment. Whereas *PER15* induction was less frequently detected at the wound site of inflorescence stems at 1 dai in ACC-treated seedlings, *ein2-1* and *etr1-3* inflorescence stems showed suberized cell layer formation similar to the wild type at 2 and 4 dai (Extended Data Fig. 10f,g,j). Considering that sealing the wound

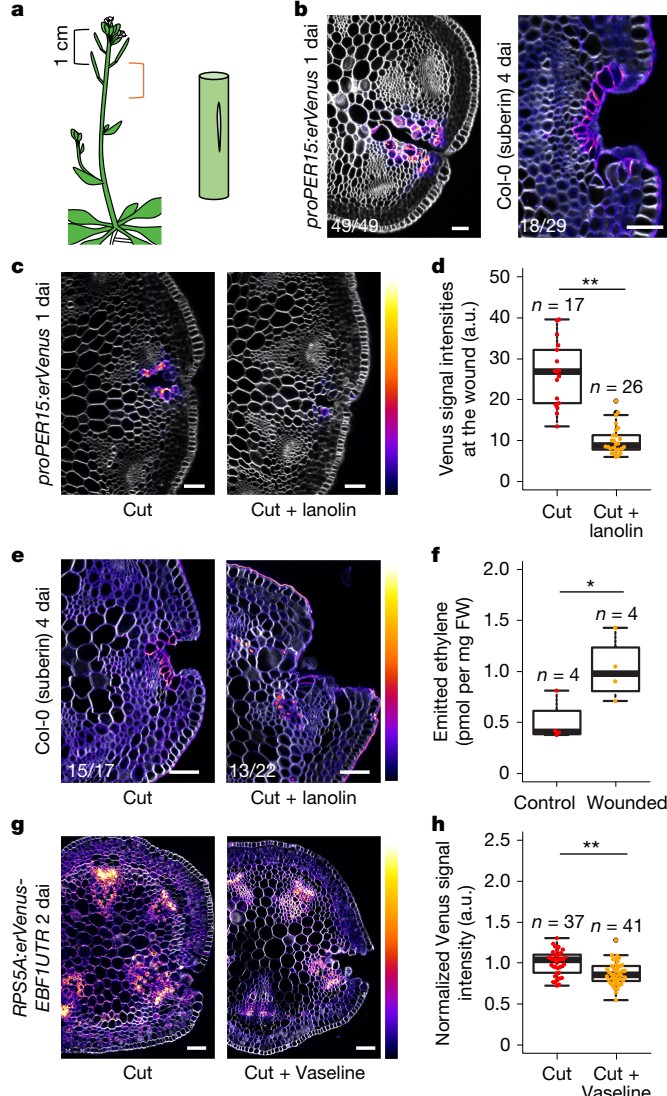

**Fig. 4 | Barrier integrity is monitored with gas-mediated surveillance system in shoot. a**, Schematic of the wounding experiment in inflorescence stems. Inflorescence stems were longitudinally cut (orange bracket region). **b**, A cross-section of *proPER15:erVenus* (left) and wild-type (right) inflorescence stem at 1 and 4 dai, respectively. **c**, Cross-sections of *proPER15:erVenus* inflorescence stems at 1 dai grown without (cut) or with (cut + lanolin) lanolin or Vaseline at the wound for 1 dai. **d**, Quantification of Venus signal intensities at the wound site in *proPER15:erVenus* inflorescence stems at 1 dai. The wound was not (cut) or was (cut + lanolin) covered with lanolin or Vaseline right after the injury. Two-tailed Wilcoxon rank-sum test was used (**$P < 0.01$). **e**, Cross-sections of wild-type inflorescence stems at 4 dai. The wound was not (cut) or was (cut + lanolin) covered with lanolin for 4 dai. **f**, Concentration of ethylene emitted from inflorescence stems 3 h after injury. The wild-type inflorescence stems were not (control) or were (wounded) injured along the longitudinal axis. Two-tailed Welch's *t*-test was used (*$P < 0.05$). **g**, *RPS5A:erVenus-EBF1UTR* signals in inflorescence stems at 2 dai. The inflorescence stems grown without (cut) or with (cut + Vaseline) Vaseline at the wound after the injury. **h**, Venus signal intensities in the cortical cells near the wound were measured and normalized with *RPS5A:erVenus-EBF1UTR* inflorescence stems grown without Vaseline. Two-tailed Wilcoxon rank-sum test was used (**$P < 0.01$). *n* indicates the number of examined cross-sections in **d**,**h** and the number of repeats in **f**. Venus signal intensities in **b**,**c**,**g** and intensity of suberin staining with fluorol yellow in **b**,**e** are shown according to the colour scales on the right. The top (brighter) area of the scale represents a higher intensity of signals. White, SR2200 (cell wall) in **b**,**c**,**e**. Fractions on the panels indicate the proportion of cross-sections showing similar expression to that in the images. Scale bars, 50 μm.

inhibited *PER15* induction more strongly than ACC treatment and that there was no observable phenotype in *ein2-1* and *etr1-3*, the diffusion of gaseous or volatile molecules other than ethylene could be necessary for barrier re-establishment in inflorescence stems. Altogether, these results support the idea that monitoring of barrier integrity through gas diffusion is also used to re-establish the barrier in the inflorescence stems.

## Discussion

Previous studies have shown that wounding-induced ethylene biosynthesis is associated with periderm regeneration[29]. Here we show that mature *Arabidopsis* roots contain ethylene, and wounding leads to its release into the environment. This results in a reduction of ethylene signalling and, consequently, initiation of periderm regeneration. In addition, we demonstrate that hypoxia signalling is reduced owing to oxygen diffusion through the wound, leading to enhanced periderm regeneration. Whereas individual manipulations of ethylene and hypoxia have a modest effect on periderm regeneration, the combination of high ethylene and constitutive hypoxia signalling almost completely abolishes periderm regeneration. Overall, we show that the periderm integrity in the mature *Arabidopsis* root is monitored by the reciprocal diffusion of oxygen and ethylene, in and out of the root, respectively. During the formation of the Casparian strip and the embryonic cuticle, barrier integrity is monitored by specific localization of peptide hormone receptors and by spatial separation of peptide processing and perception, respectively[30–33]. The gas diffusion monitoring mechanism described here, which probably does not require a spatially restricted signalling mechanism, functions as a control system to initiate periderm regeneration and later terminate the process when hypoxia and ethylene signalling levels are restored owing to regained barrier integrity. However, because gases diffuse, precise positional cues for periderm formation are unlikely to be conveyed by these two gases. It is probable that other factors, such as peptides, other phytohormones and mechanical stresses[34], operate together with the two gases to position the regenerating periderm. From this perspective, the two gases would establish a permissive environment for the other signals to enable barrier regeneration. Further studies are required to understand these details of periderm regeneration.

We showed that injuring the epidermis of the *Arabidopsis* stem results in the formation of a phellem-like layer in the exposed tissue. Even though barrier integrity in the stem also seems to be monitored by gas diffusion, ethylene and hypoxia signalling do not have major roles in this process. Similar to in the *Arabidopsis* stem, formation of the periderm or suberized cells is elicited in some fruits when there is a crack in the waxy cuticle layer in the exocarp, the outermost barrier cell type[35,36]. It would thus be interesting to study whether similar mechanisms of monitoring gas leakage are used for barrier re-establishment in different plant organs and species. Previous reports show that plants adapt their growth and development to their environment on the basis of the accumulation of ethylene in compacted soil and waterlogged conditions[6–8]. Considering this together with our findings, we propose that monitoring the accumulation or depletion of gaseous molecules inside tissues or in the surrounding environment is a common strategy for controlling plant development.

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

## Methods

### Plant materials and growth conditions

Col-0 was used as the wild type unless stated otherwise. *ein2-1*, *etr1-3*, *ate1-2;ate2-1*, *prt6-5*, *proWOX4-erYFP*, *proPXY:GUS*, *proPXY:erVenus* and *35S:EIN3-GFP* have been described previously[10,13,27,37–40]. *ate1-2;ate2-1* mutants were reciprocally backcrossed into Col-0 for three generations and transfer DNA insertions were confirmed by genotyping to obtain double-homozygous mutants.

Seeds were sown on half-strength MS (Duchefa) plates supplemented with 0.05% MES (Duchefa), 1% agar (Duchefa) and 1% sucrose (Duchefa), pH 5.8. MS medium also contained vitamins for all experiments except for ethylene measurement from roots. After incubation at 4 °C for more than 2 days, the plates were moved to the growth chamber (22 °C; 16 h light, 8 h dark). The day that the plates were placed in the growth chamber is defined as day 0. The seedlings were moved to the new MS agar plate around day 7; for the experiment in shoots, the seedlings were transferred to soil 1 or 2 weeks after germination and were grown in a greenhouse (16 h light, 8 h dark) unless stated otherwise. For examining normal periderm, we used 14- or 16-day-old seedlings as periderm is established at that age. For assessing regeneration, we usually used 17- to 21-day-old seedlings.

The Arabidopsis Genome Initiative locus codes for the genes are as follows: *ATE1*, *AT5G05700*; *ATE2*, *AT3G11240*; *EBF1*, *AT2G25490*; *EIN2*, *AT5G03280*; *EIN3*, *AT3G20770*; *ETR1*, *AT1G66340*; *PBP1*, *AT3G16420*; *PCO1*, *AT5G15120*; *PCO2*, *AT5G39890*; *PER15*, *AT2G18150*; *PER49*, *AT4G36430*; *PRT6*, *AT5G02310*; *PXY*, *AT5G61480*; *RPS5A*, *AT3G11940*; *WOX4*, *AT1G46480*.

### Cloning

From the Col-0 genome, the 643-base-pair (bp) *EBF1* 3′ UTR with its downstream 72-bp sequence (from +2780 to +3495) was amplified by PCR using H122_EBF1_URT_F and H123_EBF1_URT_R primers[18] (Supplementary Table 1), which have a 25-bp overlap with the erVenus and attL2 sequence, respectively (EBF1 3′ UTR). erVenus/pDONR plasmid (30 ng) was amplified with the EBF1 3′ UTR PCR product (600 ng) by PCR. The amplified PCR product was digested with DpnI enzyme at 37 °C for 2 h and was used for transformation into *Escherichia coli* (DH5α) by electroporation (erVenus−EBF1 3′ UTR/pDONR). RPS5A:erVenus−3AT/VD8034GW-mTurq and RPS5A:erVenus−EBF1UTR−3AT/FRm43GW were generated from RPS5A/pDONR, 3AT/pDONR, FRm43GW, VD8034GW-mTurq and erVenus/pDONR or erVenus−EBF1 3′ UTR/pDONR, respectively, by MultiSite Gateway LR clonase reaction.

From the Col-0 genome, promoter sequences of 1.1 kilobases (kb) of *PCO1* (from −958 to +198) and 2.5 kb of *PCO2* (from −2213 to +298) were amplified by PCR using H68_proPCO1_F and H69_proPCO1_R, or H70_proPCO2_F and H71_proPCO2_R primers, respectively[23] (Supplementary Table 1). The amplified PCR products were cloned into the p1R4z-pDONR vector by BP reaction (proPCO1/p1R4z-pDONR, proPCO2/p1R4z-pDONR). proPCO1:erVenus−3AT/FRm43GW and proPCO2:erVenus−3AT/FRm43GW were generated from erVenus/pDONR, 3AT/pDONR[41], FRm43GW[42], and proPCO1/p1R4z-pDONR or proPCO2/p1R4z-pDONR, respectively, by MultiSite Gateway LR clonase reaction.

The plasmids generated in this work were introduced into Col-0.

### Surgical injury of the periderm and chemical treatment

Roots within 5 mm below the root−hypocotyl junction were used for the wounding experiment unless stated otherwise. Under a dissection microscope, the shoot was pulled upwards slightly to create tension in the roots; the roots were longitudinally cut with a razor blade. As roots where the cut reached the vascular cambium region tended to form a callus-like structure instead of the wound periderm at 4 dai, we focused on sections in which the depth of the cut reached between the phloem parenchyma contacting the periderm to the phloem-side cambium

in the following analysis unless stated otherwise. For the analysis of promoter induction at the wound site, we also excluded the sections in which the cut was just at the primary phloem pole because reporter expression tended not to be induced. To peel off the periderm for the oxygen level measurement, we made a shallow cut on the surface of the roots tangentially; the cut edge at the wound site was grasped with forceps and pulled towards the root tip.

For chemical treatment, 100 mM ACC (Merck) in water, 100 mM abscisic acid (ABA; Duchefa) in ethanol, 100 mM jasmonic acid (JA; Sigma-Aldrich) in dimethylsulfoxide, 100 mM AVG (Sigma-Aldrich) in water and 100 mM AgNO₃ (Sigma-Aldrich) in water were prepared as stock solutions. Right after the injury, seedlings were moved to plates supplemented with 10 µM ACC, 10 µM JA, 10 µM ABA, 10 µM AVG or 0.5 mM AgNO₃. For ACC treatment and its mock control, the plates were sealed with surgical tape and Parafilm. In all other experiments, plates were sealed only with surgical tape. For ABA and JA treatment, the MS agar plates containing an equivalent volume of ethanol or DMSO were used as the controls. For ethylene treatment, plates were placed in a 2-l container and 600 µl of 10% ethylene gas was injected (final concentration: 30 ppm). To ensure a consistent ethylene concentration, the container was opened and ethylene was re-injected every 1 or 2 days.

For the wounding experiment in shoot, we used an inflorescence stem whose length was between 10 and 15 cm. The region 1 to 2 cm below an inflorescence meristem was used for the surgery; we longitudinally cut the stem with a razor blade under a dissection microscope. For ACC treatment in shoot, seedlings were grown on MS agar plates for around 19 days. Inflorescence stems approximately 5 mm in length were used and a region of 2 to 3 mm below the inflorescence meristem was longitudinally injured. After the injury, seedlings were transferred to MS agar plates supplemented with or without 10 µM ACC. The plates were sealed with surgical tape and Parafilm.

To seal the wound site, lanolin was mixed with the equivalent volume of milliQ water. We applied lanolin or Vaseline to the wound after the injury. We also applied lanolin or Vaseline at the wound of control samples just before sampling to ensure that lanolin and Vaseline equally affected the histological processes.

For submergence treatment, the seedlings were first submerged into MS liquid medium not supplemented with sucrose, and their roots were injured. The seedlings were placed in the growth chamber for 24 h.

For oxygen treatment with reporter lines, MS liquid medium without sucrose was aerated with ambient air or oxygen gas for more than 10 min at 20 to 22 °C. Immediately before the surgical injury, the seedlings were submerged into aerated or oxygenated MS liquid medium and their roots were injured with a razor blade. The seedlings were transferred to glass medium bottles filled with aerated or oxygenated MS liquid medium and were placed in the growth chamber for 24 h. The method for oxygen treatment with wild-type and mutants was the same, with the exceptions that the seedlings were not submerged into MS liquid medium before the injury and they were placed in the growth chamber for 4 dai.

For hypoxia treatment, plants were grown under 5% oxygen conditions using a Whitley H85 hypoxystation (Don Whitley Scientific). Seedlings were treated for 24 h.

### Oxygen measurement

Oxygen measurement was performed using O₂ microsensors as previously described with modifications[20]. We used 24- or 25-day-old wild-type roots for the measurement. Shoot and roots below the region for the measurement were fixed on a metal mesh with plastic tape and rubber bands, and the metal mesh was set in a chamber. After we inserted an O₂ microsensor (Unisense A/S) into the region 8 to 10 mm below the root−hypocotyl junction at the depth of 70 µm from the surface, the chamber was filled with aerated deionized water. Quasi-steady-state of oxygen concentration was measured under light (50–60 µeinstein) and dark conditions; we defined the quasi-steady

state as a state in which a change in oxygen concentration was less than 5% of the total concentration for at least 5 min. The microsensor was retracted from the root, and the aerated water was drained. To measure oxygen levels when periderm integrity was compromised, the metal mesh was retrieved from the chamber, and the surface of roots used for the intact root measurement was peeled off as described above. The metal mesh was again set in the chamber and the microsensor was inserted at the depth of 50 μm in the surface-removed region. The measurement was performed in the same way as described above. Using the same roots for the measurement, the respiration and photosynthetic rate should be almost identical, and the oxygen intrusion rate would be the main cause of the oxygen level changes before and after the surgery. After the measurement, the roots were retrieved to perform histological analysis.

### Ethylene measurement

We injured roots within 3 to 4 cm below the root–hypocotyl junction in 24-, 25- or 26-day-old seedlings and removed their shoot by cutting off the hypocotyl. The hypocotyl cut sites were sealed with Vaseline. Immediately, roots were moved into a vial (volume 37.5 ml) that had been partially filled with approximately 25 ml 1% agar in 1× PBS. Plants were kept in the growth chamber (22 °C; 16 h light, 8 h dark), and samples were taken at the indicated times. After detecting the level of ethylene, we measured the fresh weight of roots and the headspace of the vial. The used Vaseline weight was subtracted from the measured fresh weight, and the corrected weight was used for quantification. Around 20 seedlings were used for measurement. For inflorescence stems, regions within 10 to 15 cm below the inflorescence meristem were used for the measurement; 3- to 5-cm stems were collected, and the flower stalks were cut. The fresh weight was measured, and the top, bottom and the flower stalk cut sites were sealed with Vaseline. For wounding, the stems were cut in the longitudinal direction with a razor blade. Immediately, the inflorescence stems were placed into a vial partially filled with approximately 25 ml 1% agar in 1x PBS. Around 30 inflorescence stems were used for measurement. In both cases, to measure the amount of ethylene emitted, 1-ml samples from the headspace were collected using a syringe 3, 6 and 24 h after the vial was closed. Ethylene measurements were performed using a Clarus 480 gas chromatograph (PerkinElmer) carrying a HayeSep N (80–100 MESH) 584 column. The oven temperature was 100 °C, with the flame ionization detector temperature set to 150 °C. The flow rate of the nitrogen carrier gas was 20 ml min⁻¹. Peak area was integrated and compared against a standard curve.

### GUS staining and microtome sectioning

Roots from 0 to 3 cm below the root–hypocotyl junction with a wound were submerged in 90% acetone on ice for 30 min. The samples were washed with 50 mM sodium phosphate buffer twice and were submerged in the GUS staining solution under vacuum for 30 to 60 min. The samples were incubated with GUS staining solution (30 mM $Na_2HPO_4$, 20 mM $NaH_2PO_4$, 1.5 mM $K_4Fe$, 1.5 mM $K_3Fe$, 500 mg l⁻¹ X-Gluc, 0.1% Triton) at 37 °C until sufficient GUS signals were detected. The roots were fixed in fixation solution (50 mM sodium phosphate (pH 7.4), 4% formaldehyde, 1% glutaraldehyde) at 4 °C overnight. To observe the samples from a lateral view under a dissection microscope, we kept the samples in 70% ethanol to remove chlorophyll.

### Vibratome sectioning and staining methods

The samples were fixed in 4% paraformaldehyde in 1× PBS for 30–60 min and washed with 1× PBS twice. The samples were embedded in 4% agarose in 1× PBS and 200 μm-thick cross-sections were made using a vibratome. The cross-sections were stained in 1× PBS or ClearSee[43] supplemented with 1 μl ml⁻¹ Renaissance SCRI 2200 (SR2200; Renaissance Chemicals) to stain cell walls[44]. To stain lignin, the cross-sections were stained in ClearSee supplemented with 1 μl ml⁻¹ SR2200 and 50 μg ml⁻¹

Basic Fuchsin (Sigma-Aldrich). For suberin staining, the cross-sections were stained in ethanol supplemented with 0.01% Fluorol Yellow 088. Before observation, the cross-sections were washed with 1× PBS supplemented with 1 μl ml⁻¹ SR2200. To visualize lignin and suberin at the same time, first the cross-sections were stained in ClearSee supplemented with 1 μl ml⁻¹ SR2200 and 50 μg ml⁻¹ Basic Fuchsin overnight. Next, suberin staining was performed as described above, and before observation, the cross-sections were washed with 1×PBS supplemented with 1 μl ml⁻¹ SR2200. To observe the root tip, the samples were fixed in 4% paraformaldehyde in 1× PBS for 30 to 60 min and washed with 1× PBS twice. The samples were then stored in ClearSee supplemented with 1 μl ml⁻¹ SR2200.

### Microscopy and data analysis

A Leica 2500 microscope (Leica) was used for light microscopy images and Leica SP5 (Leica), Stellaris (Leica) and LSM880 (Zeiss) confocal laser scanning microscopes were used to detect GFP, YFP, Venus, Basic Fuchsin, Fluorol Yellow 088 and SR2200.

Confocal images were stitched by using an ImageJ plugin (Pairwise stitching)[45]. Occasionally, this resulted in the formation of empty corners in the figure panels. To visually separate the empty corners from the black background signal of the microscopy image, we filled these empty corners with white colour.

Fluorescence signal intensities at the wound site were quantified using Fiji (v1.53) and PlantSeg. We defined the wound site as a vascular tissue located within 30 μm from the wound surface for roots. For quantification in roots, the cell wall images of the wound site were extracted by using Fiji, and the extracted images were used for segmentation by PlantSeg[46]. After manual modification, the segmented images were used to count the cell number at the wound site and to measure the Venus–YFP signal intensity in each cell by Fiji. The Venus signal intensities were measured with the minimum signal threshold to exclude the background signals. The Venus-positive cells were defined by whether the signal intensity was above the threshold specific to each line. Five to six roots were used for each treatment in each experiment; a maximum of two cross-sections of the same root at different positions were used for quantification. Cross-sections with cuts reaching the vascular cambium were excluded from quantification because they tended to show unstable periderm gene induction. For Venus signal intensity quantification in intact *proPER15:erVenus* and *proPBP1:erVenus* roots in Extended Data Fig. 2f, signal intensities were measured in periderm from the quarter of the cross-section with the minimum signal threshold. For quantification of periderm reporter lines in Extended Data Fig. 1c, five distal phloem cells in the intact tissues or near the wound in each cross-section were used for Venus signal intensity measurement with the minimum signal threshold. For *RPS5A:erVenus* and *RPS5A:erVenus-EBF1UTR* quantification in Extended Data Figs. 3h,i, 5a,b and 8e, five to ten distal phloem parenchyma cells in each cross-section were used for Venus signal intensity measurement with the minimum signal threshold. At 8 and 11 h after injury or at 2 and 5 dai, distal phloem parenchyma cells near the wound were selected for signal measurement. The average of Venus signal intensities from five to ten cells was used for statistical analysis. For *proPXY:erVenus* quantification in Extended Data Fig. 1e, we circled the vascular cambium and xylem parenchyma region and measured Venus signal intensities within the circle with the minimum signal threshold to exclude the background signals. The signal intensities were normalized with those of uninjured roots in each repeat. For *proPCO1:erVenus* and *proPCO2:erVenus* signal quantification in Fig. 3b and Extended Data Figs. 6b,c and 8f, the vascular cambium, xylem parenchyma and phloem parenchyma region were circled, and we measured Venus signal intensities within the circle with the minimum signal threshold to exclude the background signals. For *RPS5A:erVenus* and *RPS5A:erVenus-EBF1UTR* quantification in Extended Data Fig. 3j,k, Venus signal intensities were measured with the minimum signal threshold and averaged from five epidermal cells in each roots. For *RPS5A:erVenus* and

*RPS5A:erVenus-EBF1UTR* quantification in Fig. 4h and Extended Data Fig. 10h,i, 10 cortical cells or 15 cortical cells near the wound in each cross-section were, respectively, used for Venus signal intensity measurement with the minimum signal threshold. For Extended Data Fig. 10i, the signal intensities were normalized with those of control inflorescence stems in each repeat. $T_2$ or $T_3$ lines were used. For *proPER15:erVenus* quantification in shoots, the wound site within 40 μm from the wound surface was selected and the Venus signal intensities were measured with the minimum signal threshold in each cross-section. For quantification of suberized cell formation in inflorescence stems in Extended Data Fig. 10j, the length of the total wound site and the region covered with suberized cells were measured and the ratio of the suberized region was calculated.

*proPXY:GUS* signals at the wound site were classified into three categories under a stereo microscope: weak when there were no or faint signals, strong when the signals were close to saturation, or intermediate when the signals were clearly visible and weaker than the strong category.

The density of suberized cells at the wound site was calculated by dividing suberized cell number by the length of the wound site using Fiji. For the quantification, the area connected to the original periderm was excluded because it was occasionally hard to distinguish which suberized cells were originated from vascular tissue. The cross-sections showing callus-like structure formation were also excluded as suberized cell formation and morphology at the wound site is affected by callus-like structure. The mature suberized cells (more than 20 μm$^2$) were counted. For Extended Data Fig. 5e, when the re-established periderm had three gaps in the suberized cell layer or three successive cell files did not show suberized cells, we defined it as a failure in suberized cell layer formation.

Data analysis was performed with MS Excel v2308, R (v2024.04.2) and Python 3.9.

## Statistics and reproducibility

In the box plots, the 25th, 50th (central value) and 75th percentile are marked with horizontal lines within the box. The ends of the whiskers indicate the maximum and minimum values within 1.5× the interquartile range from the box ends. Outliers are shown above or below the whiskers. For quantification of suberized cell formation and gene induction at the wound site or gene expression level in the vascular region, each dot corresponds to a cross-section. For ethylene concentration, each dot indicates a repeat. All experiments were repeated at least three times (three times for Figs. 1a,c–e, 2c,e, 3a,b and 4b,c,e and Extended Data Figs. 1a,b,d,f,g, 2a–d,l,m, 3a,b(AVG and AgNO$_3$),c,d, 4a,b, 5a–d,f, 6a–c,h, 7b,c,g–i, 8a–d, 9a,c–h and 10a–c,e,g; four times for Figs. 2a,g, 3d and Fig. 4f and Extended Data Figs. 2g,h–j,n, 3b(ACC) and Extended Data Fig. 10d; five times for Fig. 4g and Extended Data Figs. 4c and 5e; six times for Extended Data Figs. 6f, 7a and 10f; nine times for Fig. 3f; all are biological repeats). Exact *P* values are provided in Supplementary Table 2.

## Reporting summary

Further information on research design is available in the Nature Portfolio Reporting Summary linked to this article.

## Data availability

The data supporting the findings of this study are available within the article. Source data are provided with this paper.

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

**Acknowledgements** We thank O. Pedersen for providing the equipment for oxygen measurement; M. Herpola, M. Zaizen-Iida, K. Kainulainen, T. Puukko and A. Lamminmäki for technical assistance; L. Ragni, T. G. Andersen and S. Takada for comments and discussion; B. De Rybel, B. Wybouw, L. Ye, R. Mäkilä, S. el-Showk, T. Blomster, X. Wang, X. Zhang and Y. Helariutta for feedback on the manuscript; and the Light Microscopy Unit (University of Helsinki) for the confocal microscope equipment and technical assistance. This work was supported by grants from the EMBO Postdoctoral Fellowship (ALTF 128-2020 to H.I.; ALTF 619-2022 to L.L.P.O.), the Japan Society for the Promotion of Science (Overseas Research Fellowships to H.I.), the University of Helsinki (Doctoral Programme in Plant Science to M.L. and J.L.O.), the Research Council of Finland (grant numbers 316544 and 346141 to A.P.M., H.I., J.L.O., M.M. and M.L.; and 346140 to A.S.), the European Research Council (ERC-CoG CORKtheCAMBIA, agreement 819422 to A.P.M., H.I., J.L.O. and M.L.; ERC-CoG SynOxyS, agreement 101001320 to F.L. and V.S.), and the Biotechnology and Biological Sciences Research Council (grant numbers BB/X001059/1 to F.L. and BB/Y000226/1 to F.L. and I.A.).

**Author contributions** A.P.M. and H.I. conceived the project; A.P.M., H.I., J.L.O., L.L.P.O., V.S., I.A. and F.L. designed the experiments; H.I. performed the experiments, except for mutant analysis of intact *ate1-2;ate2-1*, which was performed by J.L.O. and H.I., gas chromatography measurement and its analysis, which was performed by I.A., V.S. and H.I., hypoxia treatment, which was performed by V.S. and H.I., and oxygen measurement and its analysis, which was performed by L.L.P.O., A.S. and H.I.; M.M. and M.L. provided unpublished materials. A.P.M. and H.I. wrote the manuscript with input from all authors.

**Funding** Open Access funding provided by University of Helsinki (including Helsinki University Central Hospital).

**Competing interests** The authors declare no competing interests.

**Additional information**
**Correspondence and requests for materials** should be addressed to Hiroyuki Iida or Ari Pekka Mähönen.

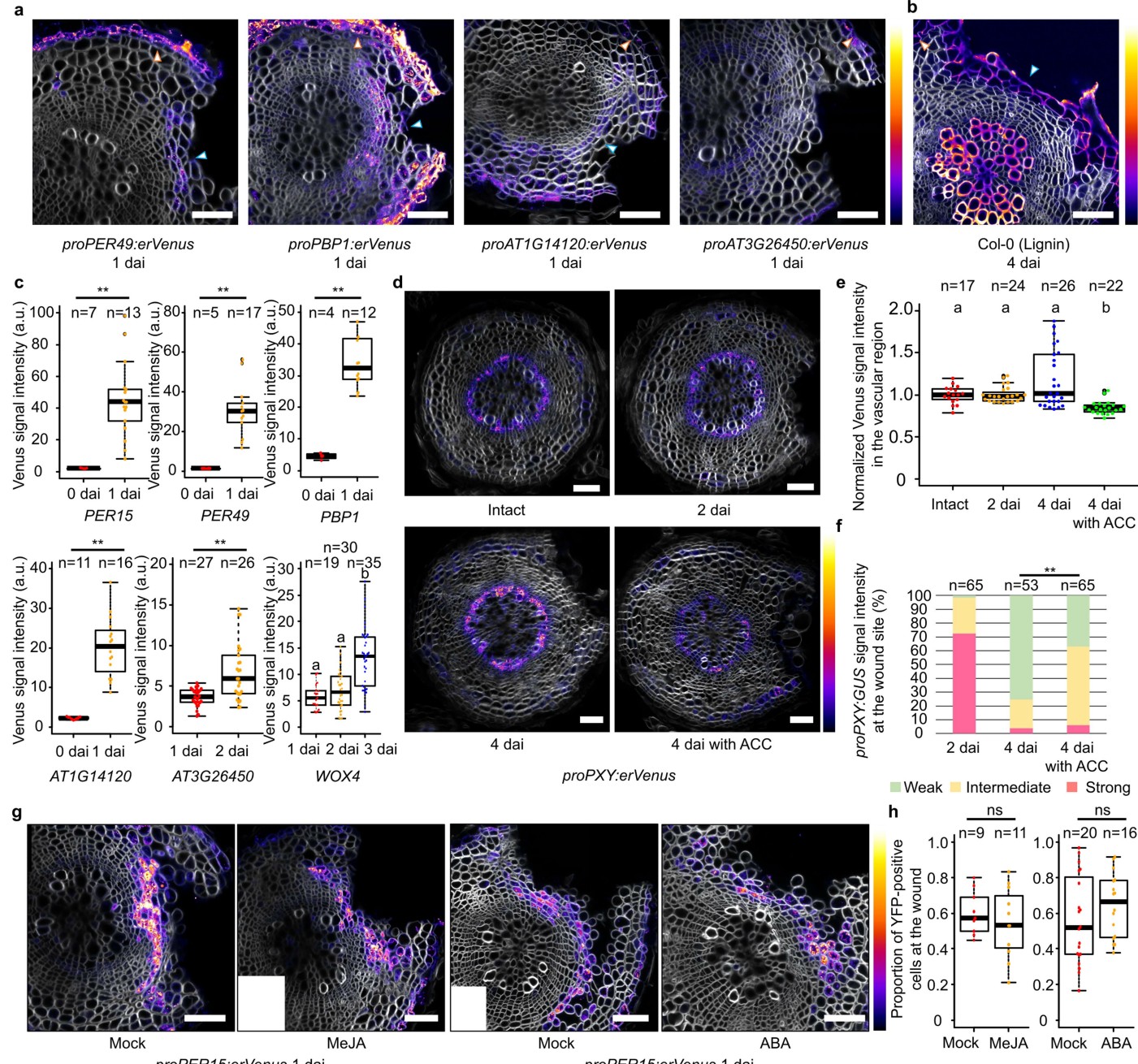

**Extended Data Fig. 1 | Gene expression, lignification and resealing the barrier during periderm regeneration. a**, Cross sections of periderm reporter roots one day after the injury. In the periderm, the promoter activity of *PER49*, *PBP1*, *AT1G14120* and *AT3G26450* was preferentially detected in the phellogen/ young phellem, in the whole periderm, in the whole periderm but weakly in the phelloderm, and in the dividing cells (presumably the phellogen), respectively. **b**, A Cross section of wild-type roots four days after 17-day-old roots were injured. Orange and blue arrowheads indicate the normal periderm or the wound sites, respectively. **c**, Venus signal intensities in distal phloem parenchyma or exposed vascular tissues of periderm reporter lines. Two-tailed Welch's *t*-test (**; *P* < 0.01) was used except for *proWOX4:erVenus*; Kruskal-Wallis test (*P* < 0.01) followed by Pairwise Two-tailed Welch's *t*-test for multiple comparisons with holm correction was used for *proWOX4:erVenus* (different characters indicate statistically significant differences between two groups; *P* < 0.05). n indicates the number of examined cross-sections. **d**, Cross sections of *proPXY:erVenus* root without injury or at 2 or 4 dai. The seedlings were grown for four days after injury on MS (4 dai) or 10 µM ACC-supplemented MS (4 dai with ACC) plates. **e**, The normalized Venus signal intensities in the vascular cambium and xylem parenchyma region of 21-day-old *proPXY:erVenus* roots without injury, and at 2 and 4 dai.

Kruskal-Wallis test (*P* < 0.01) followed by Dwass-Steel-Critchlow-Fligner pairwise comparisons was used (different characters indicate statistically significant differences between two groups, **; *P* < 0.01). n indicates the number of examined cross sections. **f**, The proportion of *proPXY:GUS* signal strength at the wound site in the seedlings at 2 and 4 dai. The *proPXY:GUS* seedlings were grown for four days after injury on MS or 10 µM ACC-supplemented MS plates. Two-sided Fisher's exact test was used to test significant difference between 4 dai and 4 dai with ACC (**; *P* < 0.01). n indicates the number of examined wound sites. **g**, Cross sections of 18-day-old *proPER15:erVenus* roots at 1 dai grown on MS plate supplemented with 0.01% DMSO (Mock), 10 µM MeJA (MeJA), 0.01% ethanol (Mock) or 10 µM ABA (ABA) for one day after the injury. **h**, The proportion of cells at the wound site showing Venus signal intensities above the threshold was quantified at 1 dai in mock, MeJA, or ABA-treated 18-day-old *proPER15:erVenus* roots. Two-tailed Wilcoxon rank-sum test was used (ns; *P* ≥ 0.05). n indicates the number of examined cross sections. Venus signal intensities in **a,d,g** and lignin staining intensities (with Basic Fuchsin) in **b** are shown according to the colour map on the right. White, SR2200 (cell wall). n indicates the number of examined cross sections. Scale bars: 50 µm. White boxes in **g** mark empty corners of stitched images. Scale bars: 50 µm.

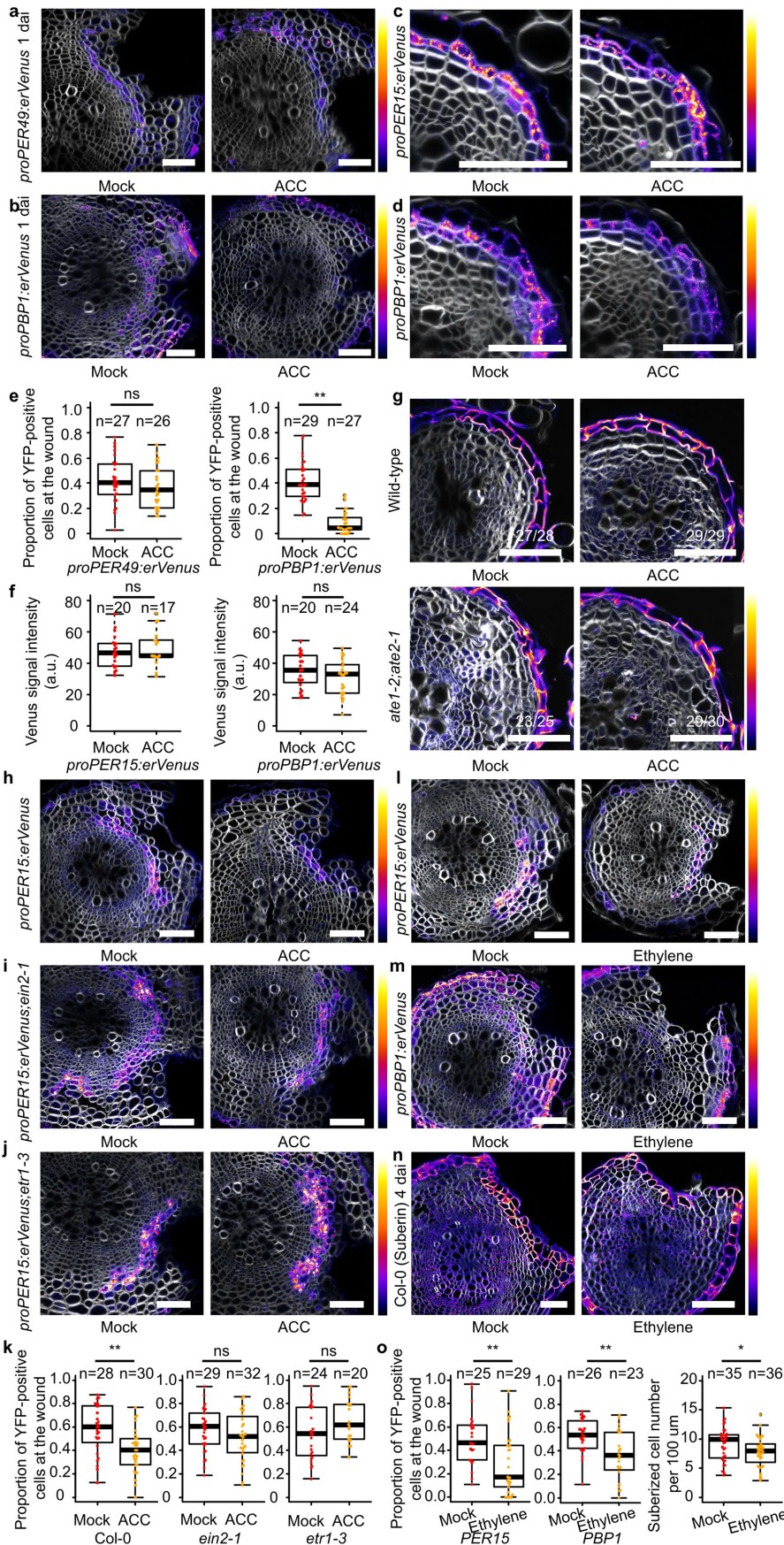

**Extended Data Fig. 2** | See next page for caption.

**Extended Data Fig. 2 | Canonical ethylene signalling pathway inhibits periderm regeneration. a,b,e**, Cross sections of 18-day-old *proPER49:erVenus* (**a**) and *proPBP1:erVenus* (**b**) roots at 1 dai grown on MS plate supplemented without (Mock) or with 10 µM ACC (ACC) for one day after the injury. The proportion of cells at the wound site showing Venus signal intensities above the threshold was quantified at 1 dai in mock or ACC-treated 18-day-old *proPER49:erVenus* and *proPBP1:erVenus* roots (**e**). Two-tailed Wilcoxon rank-sum test was used (ns; $P \geq 0.05$, **; $P < 0.01$). **c,d,f**, Cross sections of 14-day-old *proPER15:erVenus* (**c**) and *proPBP1:erVenus* (**d**) roots grown on MS plate supplemented without (Mock) or with 10 µM ACC (ACC) for two days. Venus signal intensities in periderm of *proPER15:erVenus* and *proPBP1:erVenus* without or with ACC treatment (**f**). Two-tailed Welch's *t*-test (ns; $P \geq 0.05$) was used. **g**, Cross sections of 16-day-old wild-type or *ate1-2;ate2-1* roots grown on MS plate supplemented without (Mock) or with 10 µM ACC (ACC) for four days. **h**–**k**, Cross sections of 18-day-old *proPER15:erVenus* roots at 1 dai grown on MS plate supplemented without (Mock) or with 10 µM ACC (ACC) for one day after the injury. *proPER15:erVenus* seedlings in the wild-type background were introgressed into *ein2-1* or *etr1-3* mutants, and homozygous lines for *proPER15:erVenus* transgene and mutations were used (*proPER15:erVenus;ein2-1*,

*proPER15:erVenus;etr1-3*). The proportion of cells at the wound site showing Venus signal intensities above the threshold was quantified in **k** at 1 dai in mock or ACC-treated 18-day-old *proPER15:erVenus* roots. Two-tailed Wilcoxon rank-sum test was used (ns; $P \geq 0.05$, **; $P < 0.01$). **l,m**, Cross sections of 18-day-old *proPER15:erVenus* (**l**) or *proPBP1:erVenus* (**m**) roots at 1 dai grown without (Mock) or with 30 ppm ethylene (Ethylene) for one day after the injury. **n**, Cross sections of 19-day-old wild-type roots at 4 dai grown without (Mock) or with 30 ppm ethylene (Ethylene) for four days after the injury. **o**, The proportion of cells at the wound site showing Venus signal intensities above the threshold was quantified at 1 dai in mock or ethylene-treated 18-day-old *proPER15:erVenus* (left) and *proPBP1:erVenus* (middle) roots. Two-tailed Wilcoxon rank-sum test was used (**; $P < 0.01$). The density of suberized cells at the wound site was quantified in mock or ethylene-treated 19-day-old wild-type at 4 dai (right). Two-tailed Wilcoxon rank-sum test was used (*; $P < 0.05$). n indicates the number of examined cross sections. Venus signal intensities in **a**–**d**, **h**–**j**, **l**–**n** and intensity of suberin staining with Fluorol yellow in **g,n** are shown according to the colour map on the right. Fractions on the panel indicates the proportion of cross sections showing the similar subrization as in the images. White, SR2200 (cell wall). Scale bars: 50 µm.

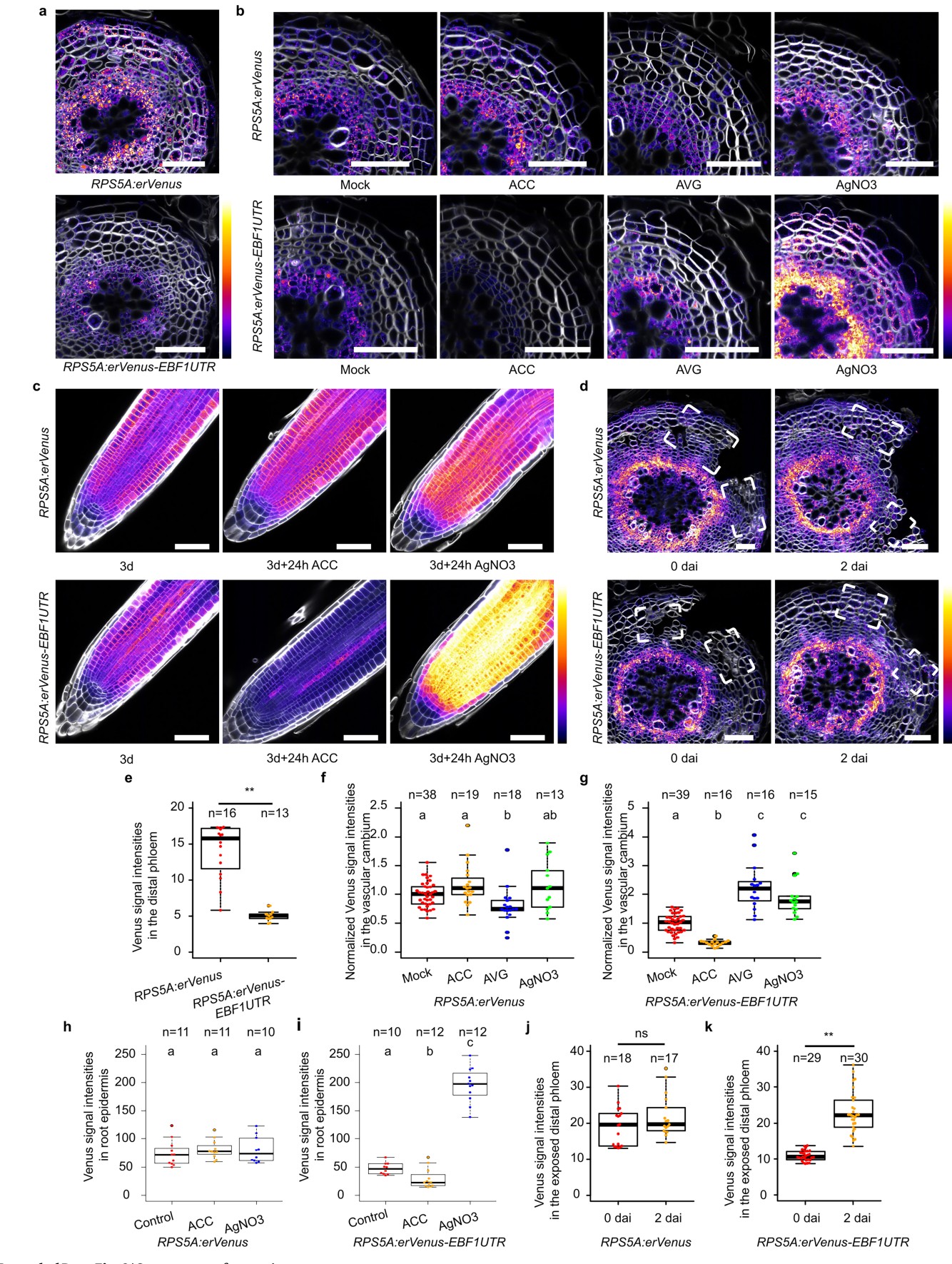

**Extended Data Fig. 3** | See next page for caption.

**Extended Data Fig. 3 | Ethylene reporter line validation and its expression after wounding. a,e**, Cross sections of 14-day-old *RPS5A:erVenus* and *RPS5A:erVenus-EBF1UTR* roots (**a**) and Venus signal intensities in distal phloem parenchyma (**e**). *RPS5A:erVenus* signals were stronger than in *RPS5A:erVenus-EBF1UTR*, especially in the periderm and phloem parenchyma indicating high ethylene signalling in these outer tissue types. **b,f,g**, Cross sections of 16-day-old *RPS5A:erVenus* and *RPS5A:erVenus-EBF1UTR* grown on MS plate (Mock), MS plate supplemented with 10 µM ACC (ACC) for two days, 10 µM AVG (AVG) or 0.5 mM AgNO3 (AgNO3) for four days (**b**). The normalized Venus signal intensities in vascular cambium (**f,g**). **c,h,i**, Confocal microscopy images of *RPS5A:erVenus* and *RPS5A:erVenus-EBF1UTR* root tip grown on MS plate supplemented without (Mock) or with 10 µM ACC (ACC) or 0.5 mM AgNO3 (AgNO3) for 24 h (**c**). Venus signal intensities in the epidermis (**h,i**). **d,j,k**, Cross sections of 19-day-old *RPS5A:erVenus* and *RPS5A:erVenus-EBF1UTR* roots right after (left) or two days after (right) injury (**d**). While *RPS5A:erVenus* signal was not changed after injury, *RPS5A:erVenus-EBF1UTR* signals were increased near the wound site, indicating reduced ethylene signalling. Venus signal intensities in distal phloem parenchyma near the wound (**j,k**). Two-tailed Welch's *t*-test (ns; $P \geq 0.05$, **; $P < 0.01$) was used for **e,j,k**. Kruskal-Wallis test ($P < 0.01$) followed by Dwass-Steel-Critchlow-Fligner pairwise comparisons (**f,g**) or Pairwise Two-tailed Welch's *t*-test for multiple comparisons with holm correction (**h,i**) was used (different characters indicate statistically significant differences between two groups; $P < 0.05$). n indicates the number of examined cross sections or root tips. Venus signal intensities are shown according to the colour map on the right. White, SR2200 (cell wall). Scale bars: 50 µm.

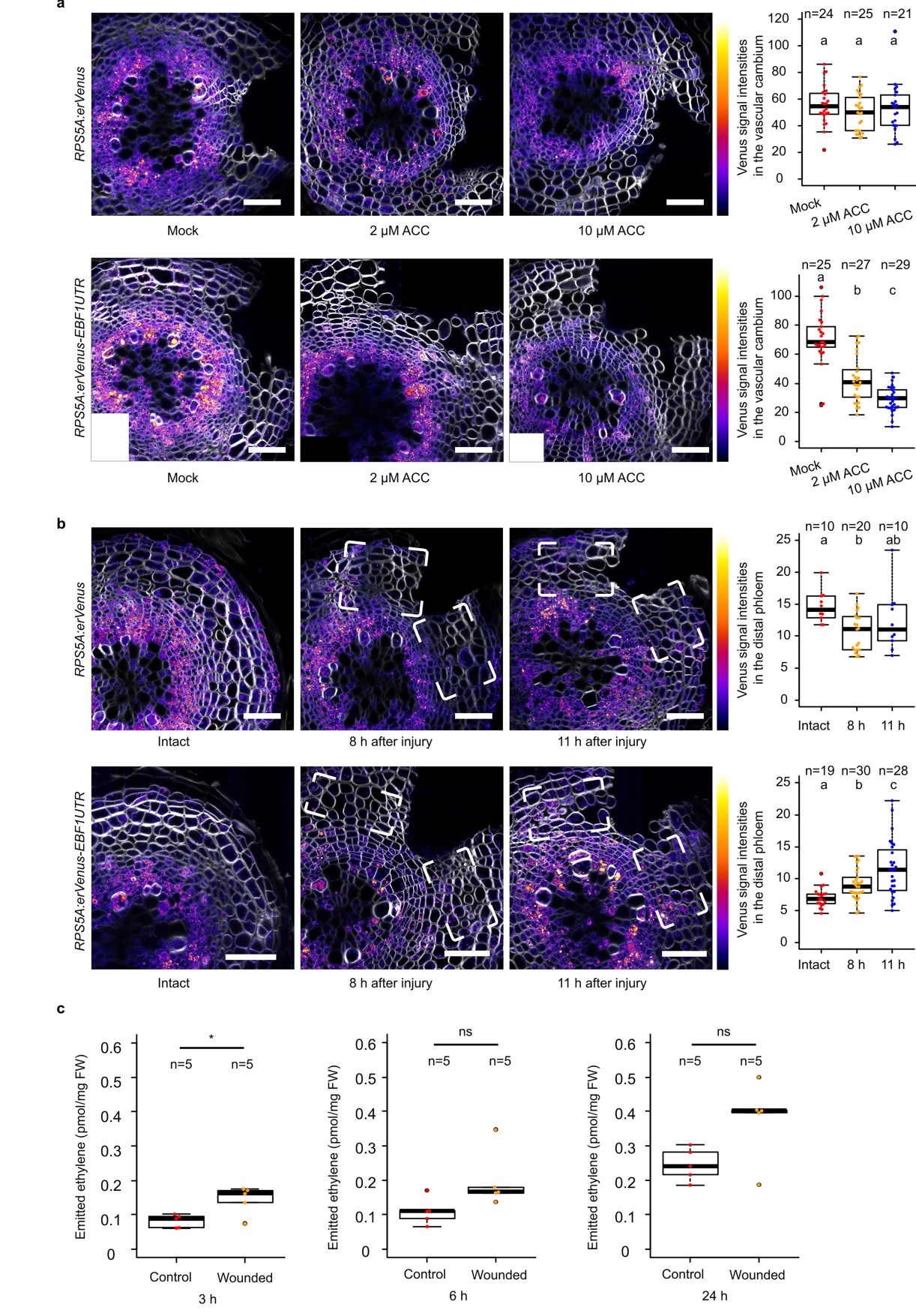

**Extended Data Fig. 4** | See next page for caption.

**Extended Data Fig. 4 | Ethylene reporter expression after ACC treatment and ethylene emission upon wounding. a**, Cross sections of 18-day-old *RPS5A: erVenus* and *RPS5A:erVenus-EBF1UTR* roots grown on MS plate supplemented without (Mock), or with 2 or 10 μM ACC for one day after the injury. Venus signal intensities were measured in the vascular cambium. *RPS5A:erVenus-EBF1UTR* signals showed a gradual decline as the ACC concentration increased, confirming that ACC treatment increases ethylene signalling level in wounded roots. **b**, Cross sections of 17-day-old *RPS5A:erVenus* and *RPS5A:erVenus-EBF1UTR* roots without injury (Intact) and 8 or 11 h after injury. Venus signal intensities were measured in the distal phloem parenchyma of intact tissues or distal phloem parenchyma near the wound. **c**, Quantification of concentration of ethylene emitted from roots. The mature part of the wild-type roots was intact (Control) or longitudinally injured (Wounded). The ethylene concentration was measured after 3, 6, and 24 h from the injury. Higher ethylene concentration was always detected from wounded roots at all time points in each replicate. Kruskal-Wallis test ($P < 0.05$) followed by Pairwise Two-tailed Welch's *t*-test for multiple comparisons with holm correction was used for **a,b** (different characters indicate statistically significant differences between two groups; $P < 0.05$). Two-tailed Welch's *t*-test was used for **c** (ns; $P \geq 0.05$, *; $P < 0.05$). n indicates the number of examined cross sections in **a,b** or the number of repeats in **c**. White boxes in **a** mark empty corners of stitched images. Venus signal intensities are shown according to the colour map on the right. White, SR2200 (cell wall). Scale bars: 50 μm.

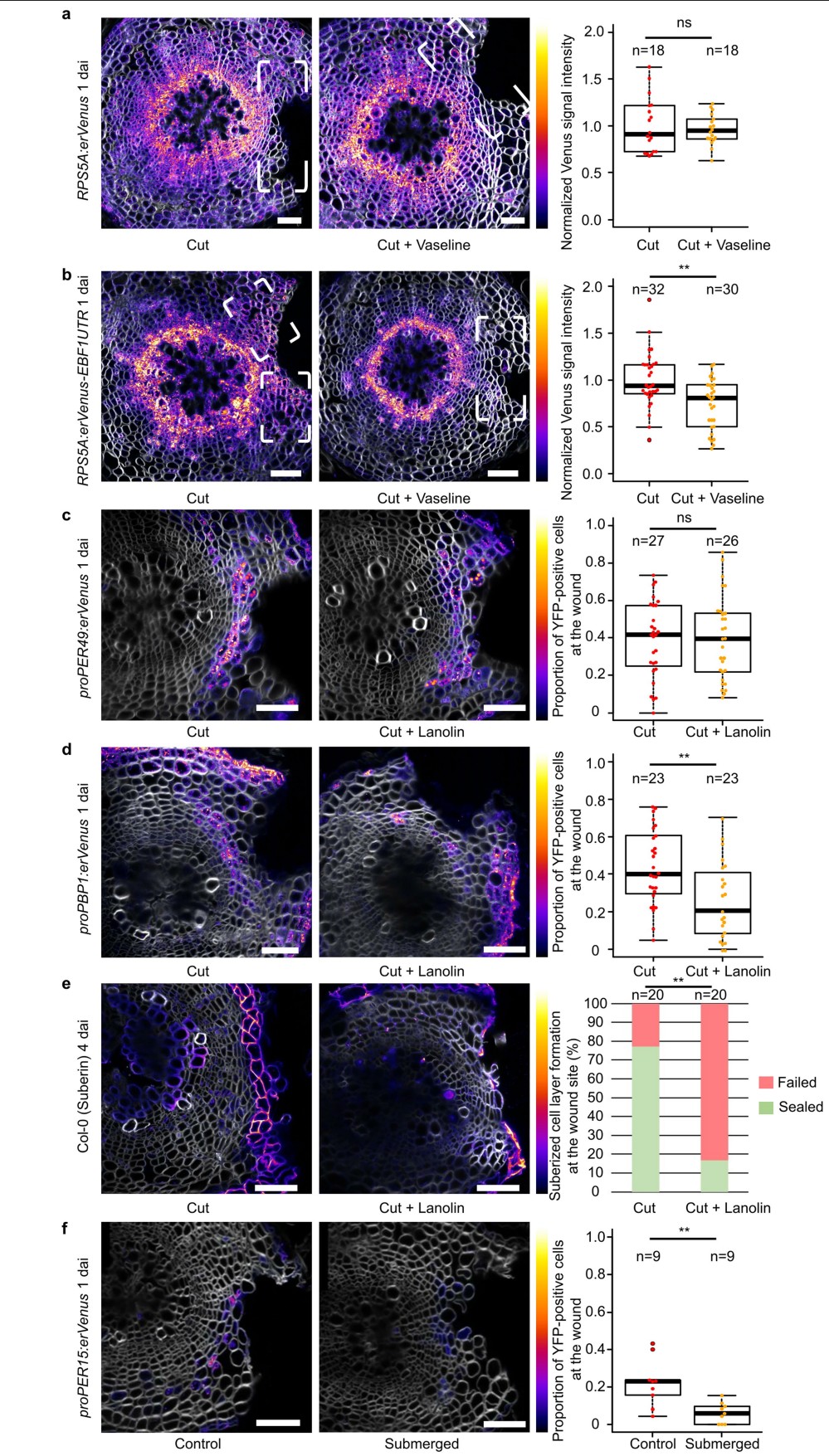

**Extended Data Fig. 5 |** See next page for caption.

**Extended Data Fig. 5 | Wound sealing maintains high ethylene signalling and inhibits periderm regeneration. a,b**, Cross sections of 18-day-old *RPS5A:erVenus* (**a**) and *RPS5A:erVenus-EBF1UTR* (**b**) roots at 1 dai grown without (Cut) or with (Cut + Vaseline) Vaseline at the wound for one day after the injury. Roots sealed with Vaseline showed reduced *erVenus-EBF1UTR* signal than unsealed roots (**b**), thus indicating higher ethylene signalling in sealed roots. No changes were observed in *RPS5A* promoter activity (**a**). Venus signal intensities in distal phloem parenchyma near the wound were measured and normalized (**c**). Two-tailed Wilcoxon rank-sum test was used (ns; $P \geq 0.05$, **; $P < 0.01$). **c,d**, Cross sections of 18-day-old *proPER49:erVenus* (**c**) and *proPBP1:erVenus* (**d**) roots at 1 dai grown without (Cut) or with (Cut + Lanolin) lanolin at the wound for one day after the injury. Proportion of cells at the wound site showing Venus signal intensities above a threshold (see methods) was quantified in 18-day-old *proPER49:erVenus* and *proPBP1:erVenus* seedlings at 1 dai. The wound was not (Cut) or was (Cut + Lanolin) covered with lanolin right after the injury. Two-tailed Wilcoxon rank-sum test was used (ns; $P \geq 0.05$, **; $P < 0.01$). **e**, Cross sections of 21-day-old wild-type roots at 4 dai. The wound was not (Cut) or was (Cut + Lanolin) covered with lanolin for four days after the injury. The proportion of cross sections showing failure in suberized cell layer formation at the wound site in 21-day-old wild-type grown without (Cut) or with (Cut + Lanolin) lanolin at 4 dai. Two-sided Fisher's exact test was used to test significant difference (**; $P < 0.01$) **f**, Cross sections of 18-day-old *proPER15:erVenus* roots at 1 dai grown in ambient air (Control) or under submergence (Submerged) for one day after the injury. The proportion of cells at the wound site showing Venus signal intensities above the threshold was quantified in 18-day-old *proPER15:erVenus* seedlings grown without (Control) or with (Submerged) submergence treatment for one day at 1 dai. Two-tailed Wilcoxon rank-sum test was used (**; $P < 0.01$). n indicates the number of examined cross sections. Venus signal intensities in **a**–**d**, **f** and intensity of suberin staining with Fluorol yellow in **e** are shown according to the colour map on the right. White, SR2200 (cell wall). Scale bars: 50 µm.

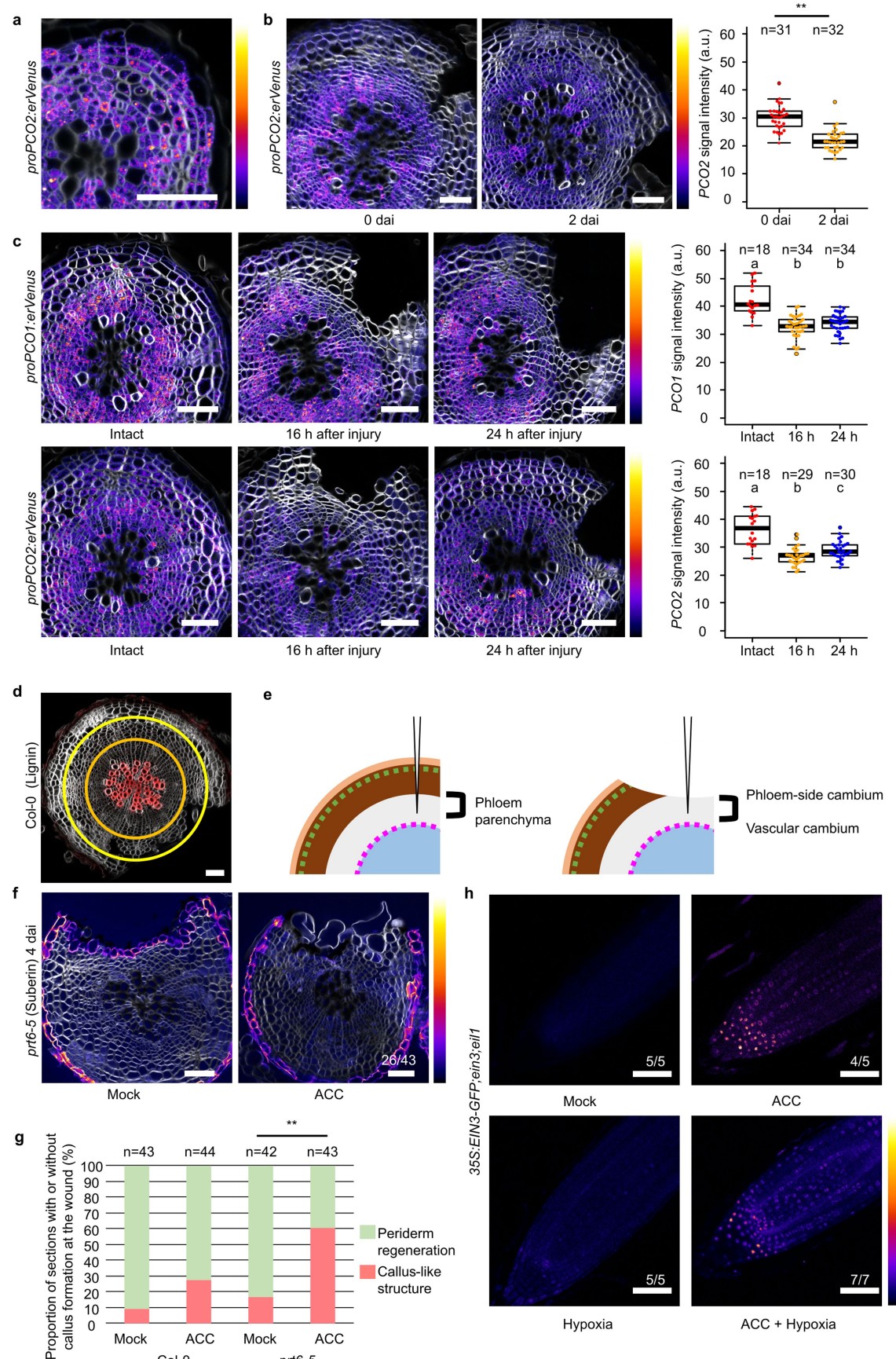

**Extended Data Fig. 6** | See next page for caption.

**Extended Data Fig. 6 | Reduction in hypoxia signalling level mediates periderm regeneration. a**, Cross sections of 14-day-old intact *proPCO2:erVenus* roots. **b**, Cross sections of 21-day-old *proPCO2:erVenus* roots right after or two days after the injury. Quantification of *proPCO2:erVenus* signal intensities in the vascular region at 0 and 2 dai. Two-tailed Welch's *t*-test was used (\*\*; $P < 0.01$). **c**, Cross sections of intact *proPCO1:erVenus* and *proPCO2:erVenus* roots and their roots 16 or 24 h after the injury. Quantification of *proPCO1:erVenus* and *proPCO2:erVenus* signal intensities in the vascular region of intact roots or at 16 and 24 h after the injury. Kruskal-Wallis test ($P < 0.05$) followed by two-sided Mann-Whitney U-test with the Bonferroni adjustment was used to test significant difference (different characters indicate statistically significant differences between two groups; $P < 0.05$). **d**, Cross section of wild-type root used for oxygen measurement. Inferred depth where the oxygen microsensor was placed before (yellow) and after (orange) removal of the periderm. **e**, Schematic drawing of the microsensor insertion site. The histological analysis suggests that the microsensor was inserted in the region around the distal phloem parenchyma cells or from the phloem-side cambium to the vascular cambium before or after the removal of the periderm, respectively. Considering that the microsensor appeared to be inserted into metabolically more active region after peeling off the periderm compared to intact root, higher oxygen level after injury indicate that the periderm prevents oxygen intrusion. **f**, Cross sections of 21-day-old *prt6-5* roots at 4 dai grown on MS (Mock) or 10 μM ACC-supplemented MS (ACC) plates for four days after the injury. *prt6* mutant phenotype was different compared to *ate1-2;ate2-1* likely due to its redundant partner, *BIG*[47]. **g**, The proportion of cross sections showing callus-like structure at the wound site in mock and ACC-treated 21-day-old wild-type and *prt6-5* roots at 4 dai. Two-sided Fisher's exact test was used to test significant difference between Mock- or ACC-treated *prt6-5* (\*\*; $P < 0.01$). **h**, The signals of *35S:EIN3-GFP* in root tips. Three-days-old *35S:EIN3-GFP* seedlings were grown on MS (Mock) or 10 μM ACC-supplemented MS (ACC) plates in ambient air or on MS (Hypoxia) or 10 μM ACC-supplemented MS (ACC + Hypoxia) plates under 5% oxygen concentration for one day. n indicates the number of examined cross sections. Fractions on the panel indicates the proportion of cross sections showing similar callus-like structure formation or root tips showing the similar expression as in the images. Venus signal intensities in **a,b,f**, GFP signal intensities in **h** and intensity of suberin staining with Fluorol yellow in **f** are shown according to the colour map on the right. Red in **d**, Basic Fuchsin. White, SR2200 (cell wall). Scale bars: 50 μm.

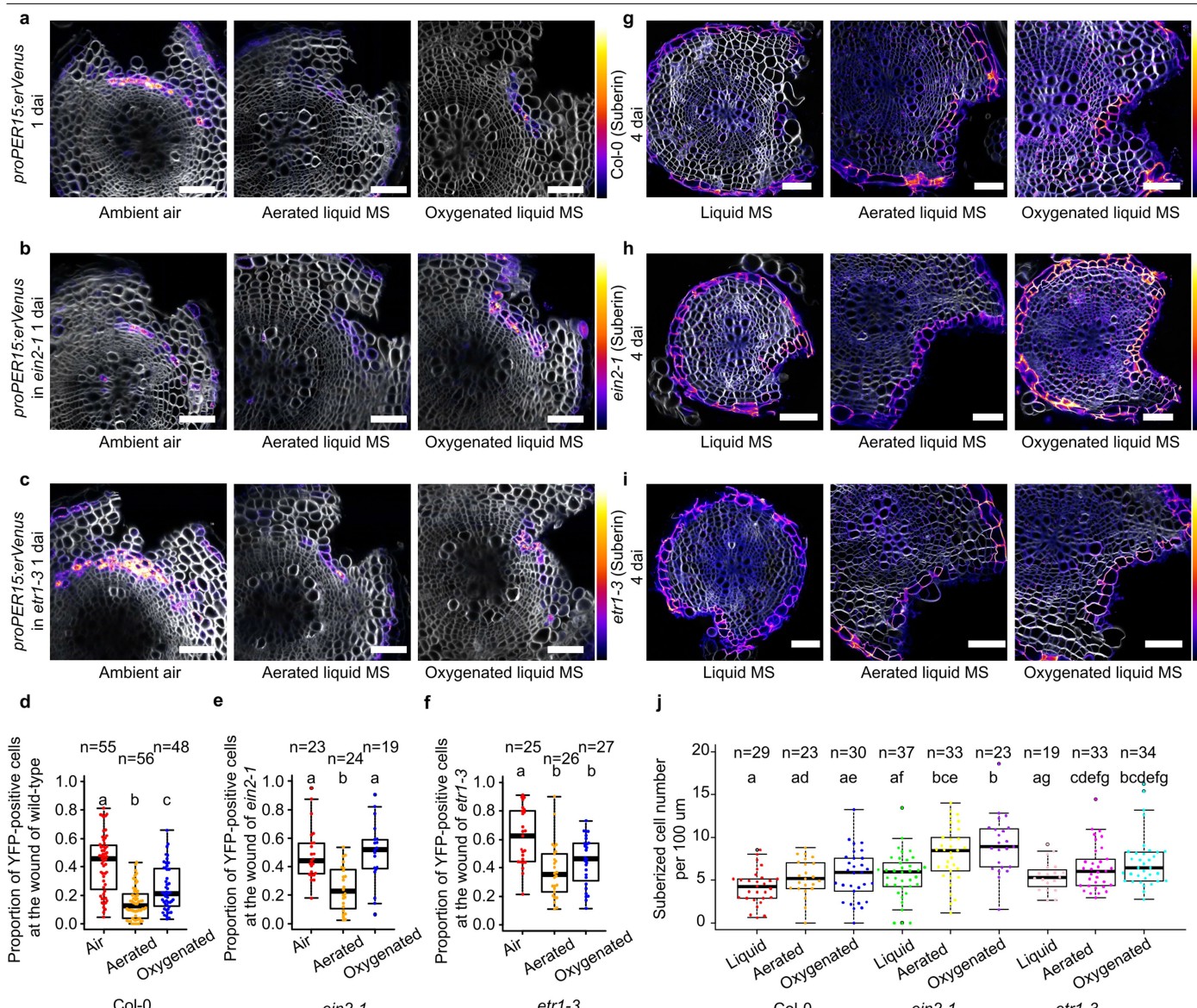

**Extended Data Fig. 7 | Lowering ethylene and hypoxia signalling rescue periderm regeneration under submergence conditions. a-c**, Cross sections of 18-day-old *proPER15:erVenus* roots in wild-type (**a**), *ein2-1* (**b**), or *etr1-3* (**c**) background at 1 dai. After the injury, the seedlings were grown on MS plates in ambient air (Ambient air, left), submerged in aerated liquid MS media (Aerated liquid MS), or oxygenated liquid MS media (Oxygenated liquid MS) for one day. **d-f**, The proportion of cells at the wound site showing Venus signal intensities above the threshold was quantified at 1 dai in 18-day-old *proPER15:erVenus* roots grown on MS plates (Ambient air), submerged in aerated liquid MS media (Aerated) or in oxygenated liquid MS media (Oxygenated). **g-i**, Cross sections of 19-day-old wild-type (**g**), *ein2-1* (**h**), or *etr1-3* (**i**) roots at 4 dai. After the injury, the seedlings were submerged in liquid MS medium without gas supplementation

(Liquid MS, left), in aerated liquid MS media (Aerated liquid MS, middle) or oxygenated liquid MS media (Oxygenated liquid MS, right) for four days. **j**, The density of suberized cells at the wound site was quantified in 19-day-old wild-type, *ein2-1*, and *etr1-3* roots at 4 dai grown in liquid MS medium without supplementing gas (Liquid), aerated liquid MS media (Aerated), or in oxygenated liquid MS media (Oxygenated) for four days. Kruskal-Wallis test followed by Dwass-Steel-Critchlow-Fligner pairwise comparisons was used (different characters indicate statistically significant differences between two groups; $P < 0.05$). n indicates the number of examined cross sections. Venus signal intensities in **a-c** or intensity of suberin staining with Fluorol yellow in **g-i** were shown according to the colour map on the right. White, SR2200 (cell wall). Scale bars: 50 μm.

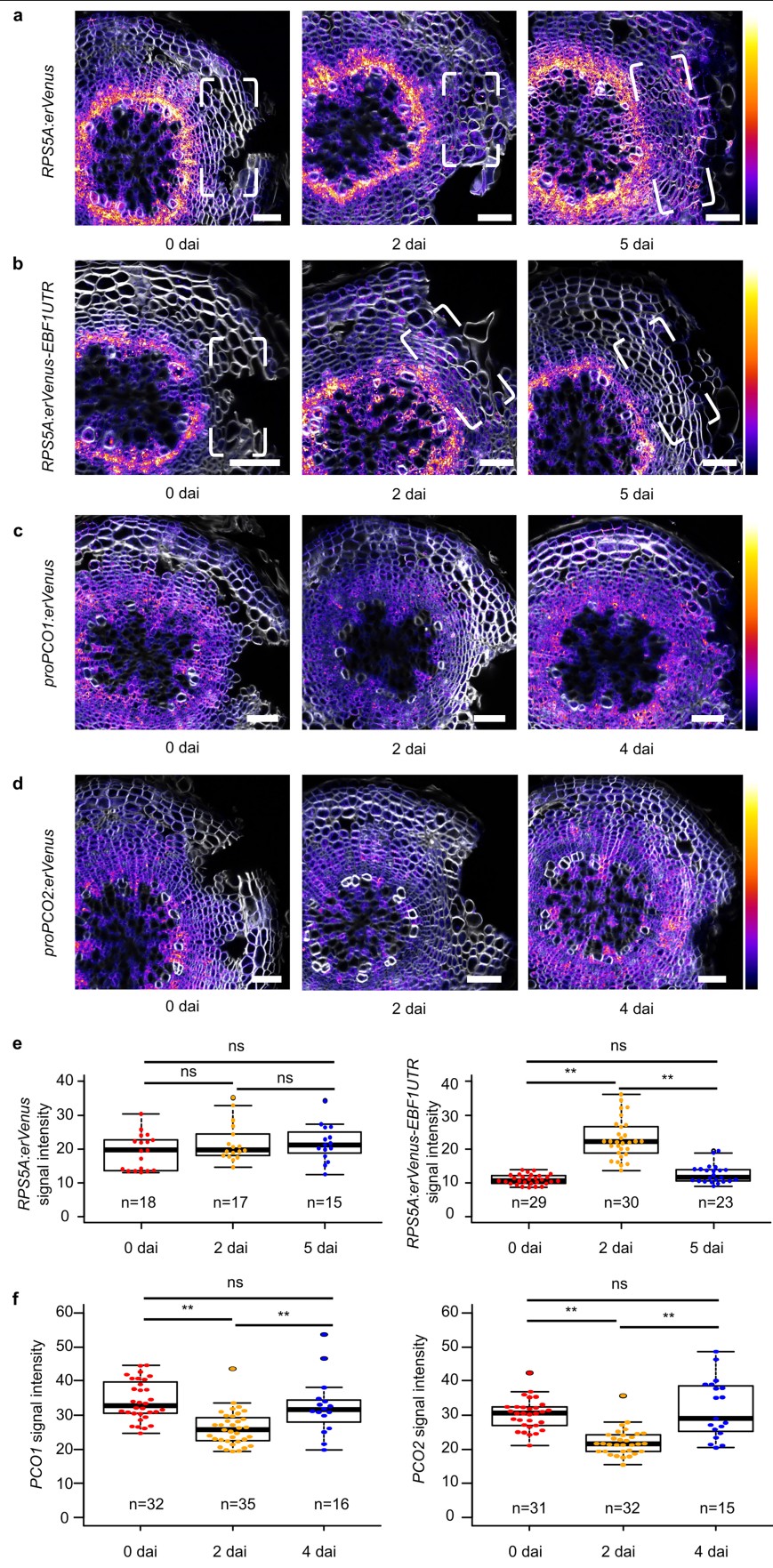

**Extended Data Fig. 8** | See next page for caption.

**Extended Data Fig. 8 | Ethylene and hypoxia signalling regain the pre-injury levels after periderm re-establishment. a,b**, Cross sections of 20-day-old *RPS5A:erVenus* (**a**) and *RPS5A:erVenus-EBF1UTR* (**b**) roots at 0 dai (left), 2 dai (middle) and 5 dai (right). *RPS5A:erVenus* signals were increased in the region beneath the regenerated periderm at 5 dai, but otherwise its expression level was not affected in the phloem parenchyma at 2 and 5 dai compared to the control. Therefore, *RPS5A:erVenus-EBF1UTR* signal intensities were examined in the phloem parenchyma neat the wound except for the area beneath the re-established periderm. **c,d**, Cross sections of 21-day-old *proPCO1:erVenus* and *proPCO2:erVenus* roots at 0 dai (left), 2 dai (middle) and 4 dai (right). **e,f**, Quantification of *RPS5A:erVenus*, *RPS5A:erVenus-EBF1UTR* signals in distal phloem parenchyma at 0, 2, 5 dai (**e**), and *proPCO1:erVenus* and *proPCO2:erVenus* signals in the vascular region at 0, 2, and 4 dai (**f**). Kruskal-Wallis test followed by two-sided Mann-Whitney U-test with the Bonferroni adjustment was used to test significant difference (ns; $P \geq 0.05$, **; $P < 0.01$). Venus signal intensities are shown according to the colour map on the right. White, SR2200 (cell wall). Scale bars: 50 µm.

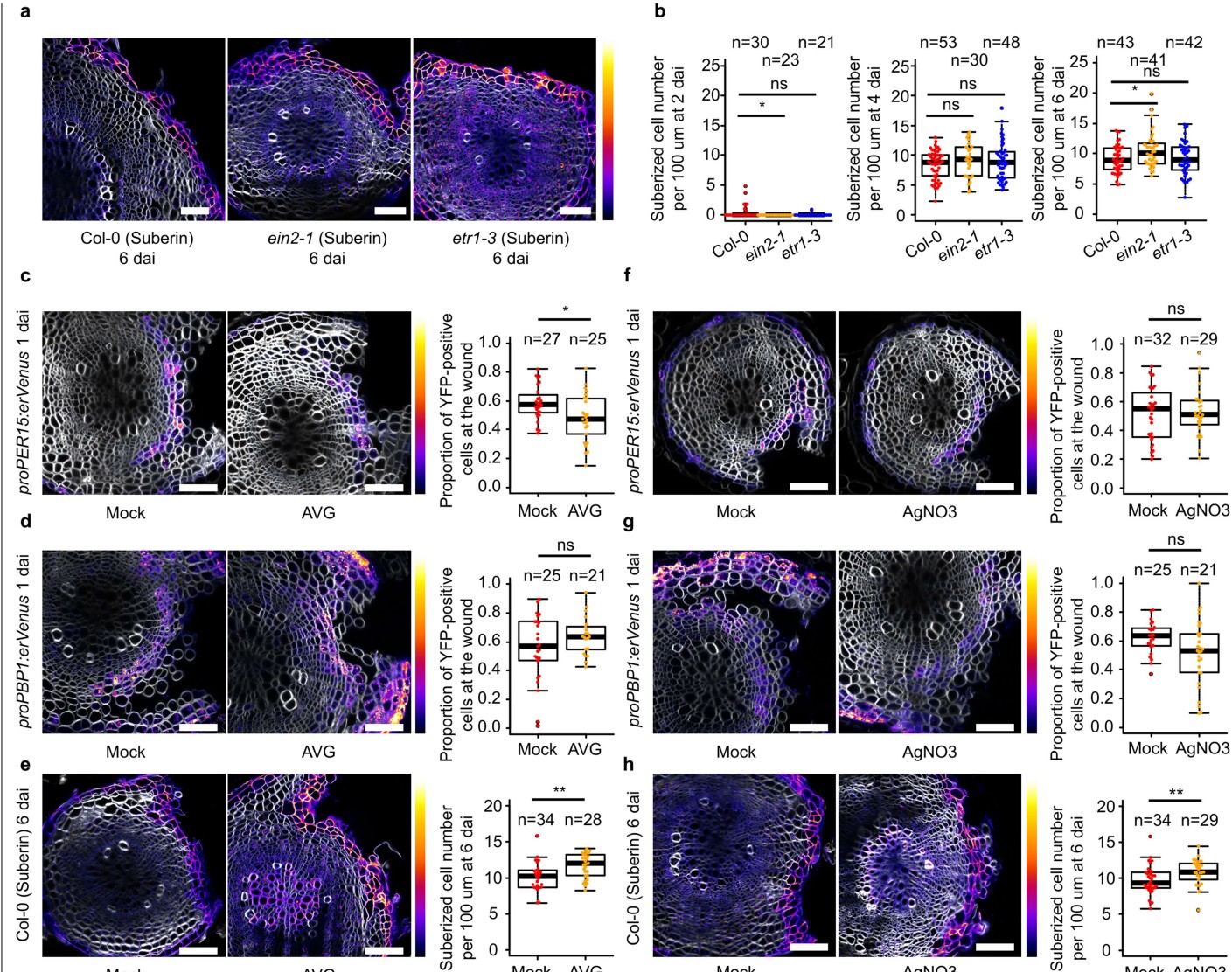

**Extended Data Fig. 9 | Reduction in ethylene signalling levels lead to increase in suberized cell formation at the wound site. a**, Cross sections of 21-day-old wild-type, *ein2-1*, or *etr1-3* roots. **b**, The density of suberized cells at the wound site was quantified in wild-type, *ein2-1*, or *etr1-3* roots 2, 4, and 6 days after 15-day-old roots were injured. Two-tailed Wilcoxon rank-sum test was used (ns; $P \geq 0.05$, *; $P < 0.05$). The difference between *ein2-1* and *etr1-3* would be due to the nature of mutations. Because *etr1-3* is a dominant negative mutation, tissues in *etr1-3* mutant still respond to ethylene when *ETR1* is not expressed. **c,d,f,g**, Cross sections of 18-day-old *proPER15:erVenus* (**c,f**) or *proPBP1:erVenus* (**d,g**) roots at 1 dai grown without (Mock) or with 10 μM AVG (AVG) (**c,d**) or 0.5 mM AgNO3 (AgNO3) (**f,g**) for one day after the injury. The proportion of cells at the

wound site showing Venus signal intensities above the threshold was quantified at 1 dai in mock, AVG or AgNO3-treated 18-day-old *proPER15:erVenus* and *proPBP1:erVenus* roots. Two-tailed Wilcoxon rank-sum test was used (ns; $P \geq 0.05$, *; $P < 0.05$). **e,h** Cross sections of 21-day-old wild-type roots at 6 dai grown without (Mock) or with 10 μM AVG (AVG) (**e**) or 0.5 mM AgNO3 (AgNO3) (**h**) for six days after the injury. The density of suberized cells at the wound site was quantified in mock, AVG or AgNO3-treated 21-day-old wild-type at 6 dai. Two-tailed Wilcoxon rank-sum test was used (**; $P < 0.01$). n indicates the number of examined cross sections. Venus signal intensities in **c,d,f,g** and intensity of suberin staining with Fluorol yellow in **a,e,h** are shown according to the colour map on the right. White, SR2200 (cell wall). Scale bars: 50 μm.

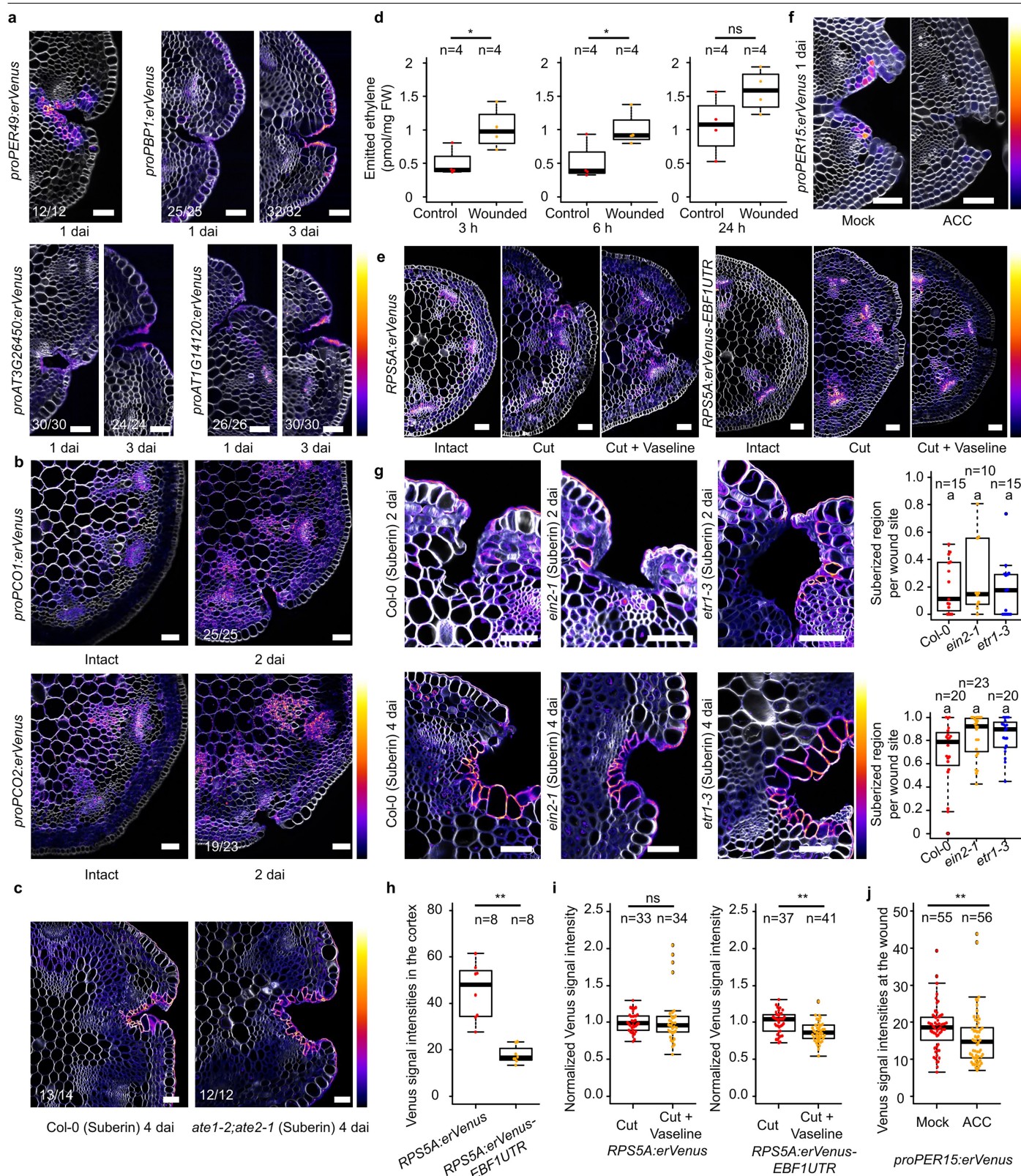

**Extended Data Fig. 10** | See next page for caption.

**Extended Data Fig. 10 | Hypoxia and ethylene signalling play a minor role in the formation of phellem-like layer in wounded inflorescence stems.**
**a**, Cross sections of *proPER49:erVenus*, *proPBP1:erVenus*, *proAT3G26450:erVenus*, and *proAT1G14120:erVenus* inflorescence stems one or three days after injury.
**b**, Cross sections of *proPCO1:erVenus* and *proPCO2:erVenus* inflorescence stems. The inflorescence stems without injury (Intact) or two days after the injury. *proPCO1:erVenus* and *proPCO2:erVenus* signals were broadly detected in their inflorescence stems with higher expression in the vascular bundle. **c**, Cross sections of wild-type and *ate1-2;ate2-1* inflorescence stems four days after injury. **d**, Quantification of concentration of ethylene emitted from inflorescence stems. The wild-type inflorescence stems were not (Control) or were (Wounded) injured along the longitudinal axis. The ethylene concentration was measured 3, 6, and 24 h after the injury. Higher ethylene concentration was always detected from wounded inflorescence stems at all time points in each replicate. Two-tailed Welch's *t*-test was used (ns; $P \geq 0.05$, *; $P < 0.05$). **e,h,i**, Cross sections of *RPS5A:erVenus* and *RPS5A:erVenus-EBF1UTR* intact inflorescence stems (Intact) or at 2 dai treated without (Cut) or with (Cut + Vaseline) Vaseline. Venus signal intensities in the cortical cells or in the cortical cells near the wound were measured in *RPS5A:erVenus* and *RPS5A:erVenus-EBF1UTR* intact or wounded inflorescence stems. Two-tailed Welch's *t*-test or two-tailed Wilcoxon rank-sum test was used (ns; $P \geq 0.05$, **; $P < 0.01$) for **h** and **i**, respectively. **f,j**, Cross sections of inflorescence stems of 21-day-old *proPER15:erVenus* seedlings at 1 dai grown on MS plate supplemented without (Mock) or with 10 μM ACC (ACC) for one day after the injury. Quantification of *proPER15:erVenus* signal intensities at 1 dai. Two-tailed Welch's *t*-test was used (**; $P < 0.01$). **g**, Cross sections of wild-type, *ein2-1* and *etr1-3* inflorescence stems two or four days after the injury. The region covered with suberized cells at the wound site was quantified. One-way ANOVA test followed by Dwass-Steel-Critchlow-Fligner pairwise comparisons was used (same characters indicate statistically not-significant differences between two groups; $P > 0.05$). n indicates the number of repeats in **d** or the number of examined cross sections in **g**–**j**. Venus signal intensities in **a,b,e,f** and intensity of suberin staining with Fluorol yellow in **c,g** are shown according to the colour map on the right. White, SR2200 (cell wall). Fractions on the panels indicate the proportion of cross-sections showing the similar expression or suberized cell formation as in the images. Scale bars, 50 μm.

# Reporting Summary

## Statistics

For all statistical analyses, confirm that the following items are present in the figure legend, table legend, main text, or Methods section.

| n/a | Confirmed | |
|---|---|---|
| ☐ | ☒ | The exact sample size (*n*) for each experimental group/condition, given as a discrete number and unit of measurement |
| ☐ | ☒ | A statement on whether measurements were taken from distinct samples or whether the same sample was measured repeatedly |
| ☐ | ☒ | The statistical test(s) used AND whether they are one- or two-sided *Only common tests should be described solely by name; describe more complex techniques in the Methods section.* |
| ☒ | ☐ | A description of all covariates tested |
| ☐ | ☒ | A description of any assumptions or corrections, such as tests of normality and adjustment for multiple comparisons |
| ☐ | ☒ | A full description of the statistical parameters including central tendency (e.g. means) or other basic estimates (e.g. regression coefficient) AND variation (e.g. standard deviation) or associated estimates of uncertainty (e.g. confidence intervals) |
| ☐ | ☒ | For null hypothesis testing, the test statistic (e.g. *F*, *t*, *r*) with confidence intervals, effect sizes, degrees of freedom and *P* value noted *Give P values as exact values whenever suitable.* |
| ☒ | ☐ | For Bayesian analysis, information on the choice of priors and Markov chain Monte Carlo settings |
| ☒ | ☐ | For hierarchical and complex designs, identification of the appropriate level for tests and full reporting of outcomes |
| ☒ | ☐ | Estimates of effect sizes (e.g. Cohen's *d*, Pearson's *r*), indicating how they were calculated |

*Our web collection on statistics for biologists contains articles on many of the points above.*

## Software and code

Policy information about availability of computer code

| Data collection | Leica LAS X |
|---|---|
| Data analysis | R v2024.04.2, Fiji v1.53, PlantSeg1.4.3, MS Excel v2308, Python 3.9 |

For manuscripts utilizing custom algorithms or software that are central to the research but not yet described in published literature, software must be made available to editors and reviewers. We strongly encourage code deposition in a community repository (e.g. GitHub). See the Nature Portfolio guidelines for submitting code & software for further information.

## Data

Policy information about availability of data

All manuscripts must include a data availability statement. This statement should provide the following information, where applicable:
- Accession codes, unique identifiers, or web links for publicly available datasets
- A description of any restrictions on data availability
- For clinical datasets or third party data, please ensure that the statement adheres to our policy

All the data that support the findings of this study are shown in this article and Extended Data Figures.

# Research involving human participants, their data, or biological material

Policy information about studies with [human participants or human data](). See also policy information about [sex, gender (identity/presentation), and sexual orientation]() and [race, ethnicity and racism]().

| | |
|---|---|
| Reporting on sex and gender | N/A |
| Reporting on race, ethnicity, or other socially relevant groupings | N/A |
| Population characteristics | N/A |
| Recruitment | N/A |
| Ethics oversight | N/A |

Note that full information on the approval of the study protocol must also be provided in the manuscript.

# Field-specific reporting

Please select the one below that is the best fit for your research. If you are not sure, read the appropriate sections before making your selection.

☒ Life sciences ☐ Behavioural & social sciences ☐ Ecological, evolutionary & environmental sciences

For a reference copy of the document with all sections, see [nature.com/documents/nr-reporting-summary-flat.pdf](nature.com/documents/nr-reporting-summary-flat.pdf)

# Life sciences study design

All studies must disclose on these points even when the disclosure is negative.

| | |
|---|---|
| Sample size | All sample size is indicated in plots. Sample-size calculation was not performed. The sizes were selected according to experiment. |
| Data exclusions | No data exclusion unless otherwise mentioned |
| Replication | All experiment is repeated at least three times. When the results were variable depending on the repeat, the experiment has been repeated more than three times. All replicates were successful. |
| Randomization | Randomization is not relevant since plants were grown in the same manner in all the experiment except for treatment. |
| Blinding | No blinding was performed. All the data was analyzed in the same manner by the same researcher(s). |

# Reporting for specific materials, systems and methods

We require information from authors about some types of materials, experimental systems and methods used in many studies. Here, indicate whether each material, system or method listed is relevant to your study. If you are not sure if a list item applies to your research, read the appropriate section before selecting a response.

### Materials & experimental systems

| n/a | Involved in the study |
|---|---|
| ☒ | ☐ Antibodies |
| ☒ | ☐ Eukaryotic cell lines |
| ☒ | ☐ Palaeontology and archaeology |
| ☒ | ☐ Animals and other organisms |
| ☒ | ☐ Clinical data |
| ☒ | ☐ Dual use research of concern |
| ☐ | ☒ Plants |

### Methods

| n/a | Involved in the study |
|---|---|
| ☒ | ☐ ChIP-seq |
| ☒ | ☐ Flow cytometry |
| ☒ | ☐ MRI-based neuroimaging |

# Dual use research of concern

Policy information about dual use research of concern

## Hazards

Could the accidental, deliberate or reckless misuse of agents or technologies generated in the work, or the application of information presented in the manuscript, pose a threat to:

| No | Yes | |
|---|---|---|
| ☒ | ☐ | Public health |
| ☒ | ☐ | National security |
| ☒ | ☐ | Crops and/or livestock |
| ☒ | ☐ | Ecosystems |
| ☒ | ☐ | Any other significant area |

## Experiments of concern

Does the work involve any of these experiments of concern:

| No | Yes | |
|---|---|---|
| ☒ | ☐ | Demonstrate how to render a vaccine ineffective |
| ☒ | ☐ | Confer resistance to therapeutically useful antibiotics or antiviral agents |
| ☒ | ☐ | Enhance the virulence of a pathogen or render a nonpathogen virulent |
| ☒ | ☐ | Increase transmissibility of a pathogen |
| ☒ | ☐ | Alter the host range of a pathogen |
| ☒ | ☐ | Enable evasion of diagnostic/detection modalities |
| ☒ | ☐ | Enable the weaponization of a biological agent or toxin |
| ☒ | ☐ | Any other potentially harmful combination of experiments and agents |

# Plants

| Seed stocks | All materials are stored in Ari Pekka Mähönen group at University of Helsinki |
|---|---|
| Novel plant genotypes | This study generated novel Arabidopsis transgenic lines in Columbia background |
| Authentication | Single insertion lines were screened among more than 12 independent transformed seedlings. T3 homozygous lines were selected based on seed coat marker or reporter gene expression. Utilized material were confirmed by genotyping, sequencing or phenotypes. |

