## [Peer Review File · Nature]

Plants monitor the integrity of their barrier by sensing gas diffusion

Corresponding Author: Professor Ari Pekka Mähönen

Version 0:

Reviewer comments:

Referee #1

(Remarks to the Author)

The ms by Lida et al. on a “A barrier-integrity mechanism in plants mediate by gas diffusion” reports on a new model injury sensing in plants that mediates regeneration. This is a novel model that has wide implications, as how the plant senses and responds to injury to embark on regeneration is a major question in our field. The development of the periderm model for regeneration studies, the novel mechanisms proposed, and the fundamental question being asked make this an impactful paper. I initially wondered how well the authors could implicate gaseous signals in the so-called barrier monitoring system, a difficult signal to work with, but I think the combination of imaging, direct gas measurements, genetics, and molecular genetic tools were skillfully employed here, and the authors got much more solid data to back up their claims than I thought they could at first. It's a very strong ms in my opinion. I have some comments on how the model is framed and then just a few lingering questions about the role of oxygen in this context.

Some of the more technical issues below relate to some misgivings about calling this a monitoring system. Perhaps I read too much into the word, but it does seem to imply that this is the first response or a primary response to injury. We know that plants respond very rapidly to injury, but the plant responses monitored in the ms were on the order of days. It is possible that many of the injury responses, even those other than patterning mechanisms, precede the responses from decreased ethylene and increased oxygen. It seems possible that the escape of ethylene and exposure to oxygen is a permissive mechanism that allows steps downstream of the original injury monitor to continue to respond. I don't think my comments diminish the importance of the model, which is primarily based on the gases being a passive mechanism that can function to sense barrier integrity independently of other potential mechanisms.

Along those lines, oxygen and reactive oxygen species have a strong influence on the cell cycle. Are the effects of increased oxygen related to a pure physiological effect on allowing cell division to proceed. Division and reprogramming may be necessary for the periderm markers to be expressed.

Also, along those lines, a body of work has shown the effects of the differential influence of hydrogen peroxide vs. superoxide on meristem growth and youth, which would certainly be affected by both the stress and increased exposure to oxygen. I am reluctant to propose anything that might resemble a fishing experiment best left for future work, but the work on ROS species in growth is so fundamental it would be important to confirm whether the authors have stumbled on a related phenomenon. There are some convenient treatments that could alter levels of H₂O₂ and superoxide post injury. e.g., can regeneration be rescued in hypoxic conditions by lowering H₂O₂ levels?

Extended Data Figure 15a seems like a critical piece of evidence as it shows a rescue by oxygen of the defective phenotype caused by ectopic ethylene signaling. Clearly oxygen is better than air for the rescue, but does air rescue over submersion without any supplemental gas? It should? After all, it is air (and presumably the oxygen level in it) that the injured plant experiences in the natural regeneration process.

I find the statistics appropriate and the ms is very clearly written.

Referee #2

(Remarks to the Author)

This MS reports roles of ethylene and oxygen in periderm formation of Arabidopsis roots and stems in response to wounding injury. Upon injury, ethylene leaks out from the wound sites and leads to reduced ethylene signaling and hypoxia signaling events. Both signaling pathways may work synergistically to regulate periderm formation. The ethylene pathway may also act in the shoot stems for periderm formation control. This study provides novel insights into the roles of ethylene and hypoxia signaling in periderm formation after injury. The following points may be addressed further.

- 1 , Fig1b, the name of each tissue should be labelled clearly on panels so readers would know easily.
- 2 , Fig1c, the structure of the cross-sections seems not consistent with the typical one in Fig1a. Each layer of tissues in panels of Fig1c is better clearly named. The red arrowhead seems not to indicate a thin cell wall position. Please check.
- 3 , Fig1d legends, please clearly indicate that the purple color in the outside cell layers represents old or newly formed periderm/phellem containing suberin? The current description in figure legends is not clear.
- 4 , Lines 85-87, the controls without ACC should be described accordingly.
- 5 , Since ethylene signaling is required to inhibit periderm formation, this connection could be further strengthened by testing whether the EIN3/EIL1 binds to the promoter regions by EMSA and inhibits promoter activity of a few periderm reporter genes (promoter-LUC assay) in Arabidopsis protoplast assay for transcriptional regulation.
- 6 , Extended data Fig1 legends. Please explain more for 'White, SR2200'.
- 7 , Extended data Fig5. The authors may want to use the ethylene constitutive response mutant *ctr1-1* to test the proPER15 expression. In this case, the expression of the marker should be reduced clearly without ACC treatment if ethylene reduced periderm marker gene expression.
- 8 , Extended data Fig8a, the difference between the two panels is not that clear and quantitation of the signals may give an accurate conclusion.
- 9 , If the ethylene signaling is involved in the periderm regeneration, the ethylene response mutants (insensitive and constitutive mutants) such as *etr1-1*, *ein2*, *ein3* (or *ein3 eil1* double mutant) and *ctr1-1* should be directly tested, just like that in Fig1d, to observe the periderm formation (faster or slower than that in Col-0) after wounding in their roots. The Extended data Fig17, and Fig21, only showed one time points. Slices at different time points may reveal the difference of time course in periderm formation. A *ctr1-1* mutant (and/or ethylene receptor loss-of-function mutants) could be used to test effect of constitutive ethylene signaling on periderm formation.
- 10 , Lines 175-177, the conclusion of 'ethylene and hypoxia signaling act synergistically' is not that solid. How would the authors differentiate between the synergistic and additive effects? A *ein2 prt6* double mutant and/or a *ein2 ate 1 ate2* triple mutants could be created to compare and quantify the effects with the single *ein2*, *prt6* or *ate1 ate2* mutants. Then the genetic interaction of the two pathways could be dissected and the conclusion may be drawn in relation to periderm formation.
- 11 , Extended data Fig14, the legend title is not accurate. Ethylene response mutants should be used for such analysis to obtain the conclusion. From the figures 14b, I would say that ACC is actually sufficient to inhibit periderm formation? Please check and revise.
- 12 , Extended data Fig15, please label the genotypes clearly on panels b, d and f, so readers would easily know the difference among the panels.
- 13 , Lines 231-237, considering that ethylene could inhibit periderm formation in both roots and stems after injury, ethylene insensitive mutants may not show strong effects. However, a *ctr1-1* mutant (and/or ethylene receptor loss-of-function mutants, or even *eto1-1*, an ethylene overproducer mutant) could be used to further test effect of constitutive ethylene signaling/overproduction on periderm formation. If these mutants show reduction of periderm formation, I think the authors could at least have a conclusion that lower ethylene levels/signaling may be required for periderm formation in both wounded roots and stems.

Referee #3

(Remarks to the Author)

This work from Lida et al. presents intriguing data sets on periderm regeneration, using older Arabidopsis root systems as a model. Based on their observations that treatment with the ethylene precursor ACC is able to suppress wound-induced periderm regeneration, the authors suggest an elegant model whereby wounding lowers ethylene levels in the tissue, through release of this gaseous hormone into the environment. This is suggested to serve as a signal for the initiation of periderm regeneration. As second signal required for regeneration is proposed to be the entry of oxygen into the hypoxic environment in the inner root tissue layers. This is a very elegant and intuitive model and the authors provide a set of well-executed experiments in support of this model, demonstrating the positive or negative effects of oxygen or hypoxia, respectively, on periderm regeneration, in combination with ethylene. The authors genetically manipulate hypoxia and ethylene signaling and provide measurements of ethylene release after wounding. They also provide data using reporter lines that can serve as proxies for oxygen status in the plant.

While I follow the authors in their assertion that they have identified a monitoring system for periderm integrity, I think they should acknowledge that the two signals – lower ethylene, increased oxygen – are clearly not sufficient to induce periderm formation/regeneration. If this were the case *ein2* ethylene signaling mutants should be "hair-triggered" to form periderm and increasing oxygen levels should lead to spontaneous/excessive periderm formation in roots, for example. This does not seem to be the case? In this context, I was surprised that the authors did not discuss their own previous paper on epidermal cell fate acquisition in any detail. In this previous paper in Nature Communications, the authors demonstrate that the acquisition of the epidermal cell fate specifier ATML1 by cortical cells is due to a release of pressure upon ablation of the overlying epidermal cells. This is based on the observation that application of pressure is sufficient to suppress ATML1 expression in cortical cells at the wound site.

Did the authors investigate whether pressure release is an additional signal that is required for periderm regeneration? I think it would be logical to present some data on this in this story.

I would also have loved to see experiments in high oxygen conditions (pure oxygen? Or air with increased oxygen levels?). This would allow to test whether oxygen increases can enhance periderm regeneration, or compensate for lower ethylene signaling, which would further strengthen their model. The authors present some data on aerated whether oxygenated 0.5 MS, but I don't find the effects terribly impressive and I wondered whether increased oxygen in a non-liquid environment would have stronger effects?

Additional comments:

- I wonder if there is any way to get around the problem that wounding induces ethylene – because of this, it is difficult to claim that enhanced ethylene diffusion across the wound is occurring, since it could simply be due to the initially enhanced ethylene production? Have the author tried a pre- or co-treatment with ethylene biosynthesis inhibitors? Like this any increase in ethylene release after wounding would be more certain to be due to enhanced diffusion?
- The authors base a lot on PER15 as a periderm marker and I wonder how specific this marker is. In Ext Data 4 they use PBP1, which also seems to work very well and I think it should be used as a second marker in some central experiments, such in Fig. 2G, 3D
- Why are the results for *etr1* in Ext. Data 15 different from *ein2*? In ext data 5 *etr1* is as strong as *ein2*.
- I think Ext Data 6 would benefit from additional quantification of the signals. It is hard to see how the authors simply classified the quantitative difference in signal strength as different categories.

Referee #4

(Remarks to the Author)

The manuscript by Lida et al. proposes that plants can monitor the integrity and regulate the repair of barrier tissue via gas diffusion. Data is presented to support three main conclusions: that periderm integrity is sensed by ethylene leaking out and oxygen entering (following wounding); that this is mediated by attenuated ethylene signaling and hypoxia signaling and this in turn mediates periderm regeneration.

The novelty here lies in linking both ethylene and oxygen dynamics to barrier formation following wounding (although the data presented is not completely convincing for the claims made— see my comments below). Also, there is previous work that has established the link between ethylene, wounding and periderm/suberization formation in sweet potato and potato (which I don't believe the authors cited). Reference: <https://doi.org/10.21273/HORTSCI.24.5.805> (St Amand and Randle, 1989); <https://doi.org/10.1016/j.postharvbio.2004.05.012> (Lulai and Suttle 2004)

Comments:

- 1) Periderm re-establishment: the authors use periderm reporter genes, phenologen reporters and suberin and lignin to demonstrate periderm recovery following injury. Suberin and lignin is very clearly visible, but the markers are harder to see and I see no quantitation for these reporter signals. This is needed.
- 2) To link ethylene function to periderm regeneration, the authors assess the induction of PER15 as a periderm marker (to find that ACC suppresses expression); Here the authors use the wounded 'mock' as controls. Since wounding itself triggers ethylene production, a better control to see 'ethylene/ACC' triggers up/downregulation of marker genes is to also treat an unwounded root (with/without the hormone treatment). This goes also for any readout (eg suberin) where ACC is added to injured roots. The response observed in these roots is likely that to a very high concentration of ethylene (wounding induced ethylene + ACC). And it is well known that ethylene like most hormones elicits dose dependent responses.
- 3) Since PER15 is primarily used throughout the manuscript as a readout for periderm regeneration, can the authors show that *per15* mutants are incapable of periderm regeneration?
- If ethylene is a negative regulator of periderm formation ethylene addition should prevent normal periderm regeneration. This is tested by addition of ACC and the finding that the number of suberized cells is lower or discontinuous. In Figure 2 a, b, I missed the data for controls (unwounded roots). Also here an important experiment is to not only add the signal (in this case ACC) to wounded roots but remove the signal (ethylene inhibitor). Adding ACC, to wounded roots that already are producing ethylene would lead to very high levels of ethylene and might confound the output (hormones can have opposite effects depending on concentration). With these experiments, one would expect to see opposite trends for periderm markers and importantly suberisation. The cleanest is a chemical approach. Ethylene mutants tend to confound effects due to their

strong developmental effects (unless these are conditional mutants). Also linked to this comment, considering that one of the main conclusions of this work is the involvement of ethylene in barrier repair, I find it very puzzling that ethylene mutants have no suberin phenotype (Extended data fig 17). This suggests that while ethylene(ACC) influences the regulation of PER15, ethylene signaling is not required for barrier regeneration. Even if hypoxia signalling is active in these mutants, one would expect some contribution of ethylene signaling. Did the authors try block hypoxia signalling? For example, by providing hyperoxia? What would happen if lanolin is applied to these mutants?

4) The functionality of the barrier tested for ACC and as shown in extended data figure 2c is not very convincing i.e. ACC treatment does not really reverse the 4 dai trend. Should the ACC treatment not be done earlier (unless I read the methods wrong and it is only the 4dai measurements that are shown).

5) In Figure 2f the EBS:GUS marker is used as a readout of ethylene signalling (and in the extended data RPS5A line). Because there seems to be a discrepancy between ethylene emitted and ethylene signaling, there should be similar time points for the two. For example ethylene emission steadily increases over time (as measured upto 1dai), but ethylene signaling as indicated by the two ethylene reporter lines suggests downregulation. In Figure 2f 0dai shows intense staining - indicating a very rapid ethylene signaling response. An important missing control here is the unwounded root, that should theoretically show no staining. Also missing are the earlier time points that were used for measuring ethylene emission.

6) I have a problem with the term 'ethylene leaks out'. This suggests a containment of the hormone under normal conditions and a breach (wounding) resulting in the hormone leaking out along a strong concentration gradient. This should then occur instantaneously within very short durations, unlike the decline in ethylene signaling seen here (also why it is important to have ethylene signaling monitored dynamically). This is assuming that under control conditions there are high levels of the hormone compared to wounded. This is not the case (as also the data here points out; or I would recommend additional measurements eg: ethylene marker genes). Unless the authors interpret the increased ethylene emission as ethylene 'leaking' out. It is well established that wounding upregulates ethylene biosynthesis (and therefore ethylene emission), which is what I interpret the data here as. The authors should check this (and untangle ethylene biosynthesis and signaling) using ethylene biosynthesis mutants.

7) I'm a bit confused between the correlation (or lack of) of ethylene emission and ethylene signaling. Can the authors (repeating a comment I made above) demonstrate using similar time points for ethylene emission and ethylene response markers/reporters? I would also like to see ethylene signalling output for the lanolin experiments. Ethylene can feedback inhibit itself, which means high ethylene levels can downregulate ethylene responses and this would reveal that.

8) Although the authors have taken care to show that ACC effects are not independent of its conversion to ethylene, I would suggest an experiment with ethylene gassing (like what they have done for hypoxia) and observe marker gene and suberisation.

9) To resolve the dose dependent effects of ethylene, the authors could do an experiment with a low and high concentration of ethylene (on wounded roots) and then assess output.

10) For both ethylene reporters (especially since one is newly synthesized reporter line and the other famously leaky), a minimum positive control is needed (plants treated with and without ethylene/with and without ethylene inhibitor).

11) The authors only look at O₂ as an additional gas, and I guess the choice was due to its recent emergence as an important developmental regulator. However there are other gases that could be physiologically relevant eg: NO, CO₂. Did they not test those, or were the results negative – could the authors comment?

12) Throughout the study the plants use vary in age between 14-21 old. Is there a reason for this choice? Plants can vary in their response depending on age.

13) Figure 3b – did the authors look at shorter time points for the PCO response? Again in keeping with gas diffusion, I would expect the effects to be very quick, i.e. within a few hours. Figure 3c, could the authors recalibrate the scale to pO₂ in %? This would help non-experts relate the concentration changes easier. Was the choice of 5% Oxygen in the hypoxia experiments guided by the oxygen levels observed in 3c?

14) The data showing that reduced hypoxia signaling is required for periderm generation is not very convincing. Difference in PCO₁ signal intensity is marginal. Two mutants are used that have constitutive hypoxia signaling. While *ate1* *ate2* show the expected results, *prt6* response is similar to wild-type. Fig 3: Why not include the imaging of *prt6-5*? And how can the difference between *ate1;ate2* and *prt6-5* be explained? Since these are in the same pathway and logically should have the same phenotype?

15) Throughout the manuscript the authors generalize the observations to attribute it to 'gas diffusion'. This should be altered to specify oxygen and ethylene, since there are other gases that were not tested here.

16) A general thought about the relevance of this mechanism in natural environments: if I understand correctly the authors pull the seedlings out of its substrate to make a cut which is then exposed to the atmosphere. In natural conditions in a soil environment, there would not be such a free exchange of gases. How relevant would 'gas diffusion' be for a plant to regulate barrier properties?

17) The last statement line 237, that “ monitoring of barrier integrity via gas diffusion” is also relevant in stems is fine, but the data actually does not reveal what these gases are. Data presented does not support a strong role for either ethylene or oxygen. The lack of PCO induction is perhaps not surprising due to the photosynthetic nature of shoot tissue; Again an important time point for 4g are the unwounded tissues.

Version 1:

Reviewer comments:

Referee #1

(Remarks to the Author)

The authors have addressed all my concerns with either new experiments or reasonable explanations addressing my concerns. I think this is a valuable manuscript and novel finding in the field of plant regeneration in terms of uncovering a new primary signal that tells the plant it is injured in order to invoke repair.

Referee #2

(Remarks to the Author)

This MS has been revised and greatly improved based on my previous comments and I have no further comments.

Referee #3

(Remarks to the Author)

The authors have addressed all my comments/concerns in this carefully revised version of the manuscript. I have no further comments.

Just one remark for the authors' consideration: If higher oxygen levels cause pleiotropic effects on plant growth, maybe this is simply due to enhanced photo-respiration and could be alleviated by simultaneously raising CO₂ concentration, such that the ratio of CO₂/O₂ remains constant?

Referee #4

(Remarks to the Author)

I would like to thank the authors for clarifying my concerns and strengthening their conclusions based on additional experimental evidence.

I do not have any further comments!

Dear Referees and Dear Editor,

We appreciate the constructive comments from the referees. We have now revised the manuscript. In summary, we have

- provided a clearer and more detailed explanation of why our findings are novel (Discussion and Response to referee #4).
- performed new experiments and found that *ein2* mutant shows a significant increase in suberized cell formation at the wound site (Extended Data Fig. 9a,b).
- provided new data showing that suberized cell formation at the wound site is promoted when ethylene signalling level is downregulated using ethylene biosynthesis and signalling inhibitors (Extended Data Fig. 9e,h).
- performed a new experiment to show that, when submerged in liquid medium supplemented with oxygen, ethylene insensitive mutants, but not wild-type, show recovery in suberized cell formation (Extended Data Fig. 7g–j).
- provided a new result that a reduction in ethylene signalling level in the wound site is detected already 8 hours after injury, which is consistent with rapid ethylene release detected by GC (Extended Data Fig. 4b).
- added several quantification data as a response to various comments from the referees (e.g. Extended Data Fig. 1c, 5a,c)
- revised sections of the main text, methods, and figure legends, and made modifications to the figures to enhance the manuscript's clarity and ensure it complies with the journal's style guidelines. This includes merging the previous 21 Extended Data Figures and new figure panels into 10 Extended Data Figures.

Please find below detailed response to the referee's comments.

Referee #1 (Remarks to the Author):

The ms by Lida et al. on a "A barrier-integrity mechanism in plants mediate by gas diffusion" reports on a new model injury sensing in plants that mediates regeneration. This is a novel model that has wide implications, as how the plant senses and responds to injury to embark on regeneration is a major question in our field. The development of the periderm model for regeneration studies, the novel mechanisms proposed, and the fundamental question being asked make this an impactful paper. I initially wondered how well the authors could implicate gaseous signals in the so-called barrier monitoring system, a difficult signal to work with, but I think the combination of imaging, direct gas measurements, genetics, and molecular genetic tools were skillfully employed here, and the authors got much more solid data to back up their claims than I thought they could at first. It's a very strong ms in my opinion. I have some comments on how the model is framed and then just a few lingering questions about the role of oxygen in this context.

Authors: Thank you for the encouraging words.

Some of the more technical issues below relate to some misgivings about calling this a monitoring system. Perhaps I read too much into the word, but it does seem to imply that this is the first response or a primary response to injury. We know that plants respond very rapidly to injury, but the plant responses monitored in the ms were on the order of days. It is possible that many of the injury responses, even those other than patterning mechanisms, precede the responses from decreased ethylene and increased oxygen. It seems possible that the escape of ethylene and exposure to oxygen is a permissive mechanism that allows steps downstream of the original injury monitor to continue to respond. I don't think my comments diminish the importance of the model, which is primarily based on the gases being a passive mechanism that can function to sense barrier integrity independently of other potential mechanisms.

Authors: Thank you for this relevant comment. As the referee pointed out, it is likely that additional signals contribute to wound suberization. We do not exclude the possibility, and it does not detract the impact from our findings. However, our data indicate that gas sensing (monitoring) does play an early role in this. Indeed, while the reporter expression and morphological changes occur one to four days after the injury, we detected changes in gas levels within hours: high ethylene emission is detected three hours after the injury and oxygenation occurs within one hour. Therefore, we believe that gas diffusion is one of the primary signals to monitor the injury and barrier integrity. While it is possible, if not likely, that additional, even earlier wounding signals (e.g. Ca^{2+} , ROS) play a role in periderm regeneration, it is unlikely that ethylene or hypoxia signaling act downstream of these hypothetical signals. This is because the proposed gas diffusion mechanism does not rely on any cue other than the wound itself, through which gases can diffuse and change their concentration. However, we do agree with the referee's idea that diffusion of ethylene and oxygen acts as a permissive signal for regeneration. Also, it is hard to believe that changed gas concentrations would allow formation of such sharp concentration gradients therefore enabling spatially precise barrier formation. We have now added text in the Discussion to speculate on the permissive role of the two gases (Line 280-281). In summary, we agree with the referee's idea that diffusion of ethylene and oxygen might act as a permissive signal, integrated with other wounding-related stimuli, for regeneration.

Along those lines, oxygen and reactive oxygen species have a strong influence on the cell cycle. Are the effects of increased oxygen related to a pure physiological effect on allowing cell division to proceed. Division and reprogramming may be necessary for the periderm markers to be expressed.

Authors: Our data show that genes induced at the early stage of periderm regeneration do not require cell division. This is because early induced genes (*PER15*, *PER49*, and *PBP1*) were induced before the first cell divisions occurred during regeneration (Fig. 1c and Extended Data Fig. 1a).

SENTENCES REDACTED

FIGURE REDACTED

Also, along those lines, a body of work has shown the effects of the differential influence of hydrogen peroxide vs. superoxide on meristem growth and youth, which would certainly be affected by both the stress and increased exposure to oxygen. I am reluctant to propose anything that might resemble a fishing experiment best left for future work, but the work on ROS species in growth is so fundamental it would be important to confirm whether the authors have stumbled on a related phenomenon. There are some convenient treatments that could alter levels of H₂O₂ and superoxide post injury. e.g., can regeneration be rescued in hypoxic conditions by lowering H₂O₂ levels?

Authors: We agree that it is intriguing to explore the involvement of ROS signals.

SENTENCES REDACTED

FIGURE REDACTED

Extended Data Figure 15a seems like a critical piece of evidence as it shows a rescue by oxygen of the defective phenotype caused by ectopic ethylene signaling. Clearly oxygen is better than air for the rescue, but does air rescue over submersion without any supplemental gas? It should? After all, it is air (and presumably the oxygen level in it) that the injured plant experiences in the natural regeneration process.

Authors: We made new experiments to examine suberized cell formation under submergence (1) without aeration, (2) with aeration, or (3) with oxygenation in wild-type, *ein2*, and *etr1-3* mutant (Extended Data Fig. 7g–j, Line 203–208). We found that indeed ethylene insensitive mutants showed recovered suberized cell formation under aerated liquid medium compared to that in wild-type under liquid medium without gas supplementation. These data further support the role of oxygen in periderm regeneration.

I find the statistics appropriate and the ms is very clearly written.

Authors: We very much appreciate this encouraging comment.

Referee #2 (Remarks to the Author):

This MS reports roles of ethylene and oxygen in periderm formation of Arabidopsis roots and stems in response to wounding injury. Upon injury, ethylene leaks out from the wound sites and leads to reduced ethylene signaling and hypoxia signaling events. Both signaling pathways may work synergistically to regulate periderm formation. The ethylene pathway may also acts in the shoot stems for periderm formation control. This study provides novel insights into the roles of ethylene and hypoxia signaling in periderm formation after injury. The following points may be addressed further.

- 1, Fig1b, the name of each tissue should be labelled clearly on panels so readers would know easily.

Authors: Thank you for your feedback to improve the figure. We now mark the tissue types in the panel (Fig. 1b).

- 2, Fig1c, the structure of the cross-sections seems not consistent with the typical one in Fig1a. Each layer of tissues in panels of Fig1c is better clearly named. The red arrowhead seems not to indicate a thin cell wall position. Please check.

Authors: We now indicated the tissues in Fig. 1c (PL; phellem, PG; phellogen, PD; phelloderm) in the intact periderm. Since during regeneration the tissue structure is quite different from Fig. 1a or intact tissue, we cannot name the three cell types in the region where regenerating occurs. We thank the referee's comment about thin cell wall. The indicated cell wall is thin, so it was not clearly visible. We increased the signal intensities of the dye and increased the magnification in the insets, so that it is now easier to see the cell wall.

- 3, Fig1d legends, please clearly indicate that the purple color in the outside cell layers represents old or newly formed periderm/phellem containing suberin? The current description in figure legends is not clear.

Authors: We thank the referee to point out the missing description. We added the description in the legend "Intensity of suberin staining with Fluorol yellow is shown according to the colour map on the right."

- 4, Lines 85-87, the controls without ACC should be described accordingly.

Authors: We added the description of the control in Line 90-91 "While suberized cells in control roots formed continuous layer at the wound site, treatment with ACC after injury occasionally resulted in discontinuous suberized cell layers or callus-like structures at the wound site (Fig. 2c, d)"

- 5, Since ethylene signaling is required to inhibit periderm formation, this connection could be further strengthened by testing whether the EIN3/EIL1 binds to the promoter regions by EMSA and inhibits promoter activity of a few periderm reporter genes (promoter-LUC assay) in Arabidopsis protoplast assay for transcriptional regulation.

Authors: EMSA and LUC assay are good assays when periderm reporter genes are expected to be direct targets of EIN3/EIL1. However, we don't expect that they are the direct targets, since according to DAP-seq data, EIN3 and EIL1 binding sites were not enriched in the 5' upstream regions of *PER15* and *PBP1* (our unpublished observation in the dataset of O'Malley et al Cell 2016 (<https://doi.org/10.1016/j.cell.2016.04.038>)). Supporting this, we can observe upregulation of *PER15* and *PBP1* only 16-24h after injury (Fig. 1 and Extended Data Fig. 1a,c). Our paper is the first to report the reciprocal gas diffusion mechanism during regeneration and the genetic framework that regulates it. We think that the exact signaling steps within this framework will be studied in the future by us or other labs.

6, Extended data Fig1 legends. Please explain more for 'White, SR2200'.

Authors: We modified as following "White, SR2200 (cell wall)" in all relevant legends. Additionally, we added in the Methods "... supplemented with 1 μ l/ml Renaissance SCRI 2200 (SR2200; Renaissance Chemicals) to stain cell walls⁴⁴."

8, Extended data Fig8a, the difference between the two panels is not that clear and quantitation of the signals may give an accurate conclusion.

Authors: We have now quantified RPS5A:erVenus and RPS5A:erVenus-EBF1UTR signal intensities in distal phloem region, and this is now presented in the new Extended Data Fig. 5a,b. We have now also marked the area within phloem for quantification.

10, Lines 175-177, the conclusion of 'ethylene and hypoxia signaling act synergistically' is not that solid. How would the authors differentiate between the synergistic and additive effects? A *ein2 prt6* double mutant and/or a *ein2 ate1 ate2* triple mutants could be created to compare and quantify the effects with the single *ein2*, *prt6* or *ate1 ate2* mutants. Then the genetic interaction of the two pathways could be dissected and the conclusion may be drawn in relation to periderm formation.

Authors: We agree that we should have described the interaction as additive, not synergistic: ACC treatment in the *prt6* or *ate1;ate2* backgrounds leads to a phenotype that is additively stronger than ACC treatment or the hypoxia mutants alone (Fig. 2c, 3d, Extended Data Fig. 6e,f). We have now corrected the text accordingly in Lines 164, 174, 183, 187, and 193. To examine the role of ethylene and hypoxia signals, it would be intriguing to combine the ethylene insensitive mutant with the mutants lacking hypoxia signaling, not mutants with constant hypoxia. However, there are only three known target regulators of N-degron pathway (ERFVII, ZPR2, and VRN2) which has been shown to mediate hypoxia signalling in other developmental contexts.

SENTENCES REDACTED

Arabidopsis genome encodes other potential target genes (approximately 250 genes), and since the list of potential targets is so long, we consider the exploration of these regulators as a very interesting future study.

FIGURE REDACTED

11, Extended data Fig14, the legend title is not accurate. Ethylene response mutants should be used for such analysis to obtain the conclusion. From the figures 14b, I would say that ACC is actually sufficient to inhibit periderm formation? Please check and revise.

Authors: We have remade the analysis for this figure (new Extended Data Fig. 2g). This is because in the original experiment we used 7-day-old seedlings for ACC treatment, in which cork layer has not been fully established yet. Thus, growth retardation caused by ACC treatment made them look like periderm formation is inhibited. In revised experiment, we used 12-day-old roots which have already established cork layer and treated them with ACC for four days. We could not detect a significant defect in cork layer development, supporting that high ethylene level did not disrupt the normal periderm formation. We merged the original Extended Data Fig. 14 with other figures, so the inaccurate legend title has been removed.

12, Extended data Fig15, please label the genotypes clearly on panels b, d and f, so readers would easily know the difference among the panels.

Authors: Thank you for this suggestion. We improved the Figure by adding genotypes in the Y axis title and the bottom of the plots (new Extended Data Fig. 7d, e, f).

7, Extended data Fig5. The authors may want to use the ethylene constitutive response mutant *ctr1-1* to test the proPER15 expression. In this case, the expression of the marker should be reduced clearly without ACC treatment if ethylene reduced periderm marker gene expression.

9, If the ethylene signaling is involved in the periderm regeneration, the ethylene response mutants (insensitive and constitutive mutants) such as *etr1-1*, *ein2*, *ein3* (or *ein3 eil1* double mutant) and *ctr1-1* should be directly tested, just like that in Fig1d, to observe the periderm formation (faster or slower than that in Col-0) after wounding in their roots. The Extended data Fig17, and Fig21, only showed one time points. Slices at different time points may reveal the difference of time course in periderm formation. A *ctr1-1* mutant (and/or ethylene receptor loss-of-function mutants) could be used to test effect of constitutive ethylene signaling on periderm formation.

13, Lines 231-237, considering that ethylene could inhibit periderm formation in both roots and stems after injury, ethylene insensitive mutants may not show strong effects. However, a *ctr1-1* mutant (and/or ethylene receptor loss-of-function mutants, or even *eto1-1*, an ethylene overproducer mutant) could be used to further test effect of constitutive ethylene signaling/overproduction on periderm formation. If these mutants show reduction of periderm formation, I think the authors could at least have a conclusion that lower ethylene levels/signaling may be required for periderm formation in both wounded roots and stems.

Authors: We have combined the response to these three related comments here.

We have now provided new data on suberin formation in *ein2* and *etr1-3* roots and stems at 2 or 6 days after injury. Our new data demonstrate that *ein2* roots showed a significant increase in suberized cell number at the wound site 6 days after the injury (Extended Data Fig. 9a,b, Line 223-224). Previously, we did not observe significant difference 4 days after injury when the fully sealed barrier is just completed (Extended Data Fig. 9b). Thus, this finding supports the idea that ethylene level increase after sealing is required to precisely terminate the suberization process. Furthermore, we also see significant difference in suberization of *ein2* and *etr1-3* compared to wild-type when submerged in oxygenated liquid MS media (Extended Data Fig. 7g-j, Line 203-208). Additionally, as referee 4 suggested, we have now used ethylene biosynthesis or signalling inhibitors and found that these chemical treatments enhanced suberized cell formation, further supporting the role of ethylene signalling in periderm regeneration (Extended Data Fig. 9e,h, Line 230-231). Please refer to our response to comment 4 of referee 4, in detail.

The referee suggested us to study periderm regeneration in *ctr1* mutant. However, due to the constitutive ethylene signaling, *ctr1* mutant is a small and sick plant (images below on the left from Cheng et al 2009 (<https://doi.org/10.1007/s11103-009-9509-7>)). We also examined their secondary tissues and found severe developmental defects. At the typical age for analysis (14 days), *ctr1-3* has not yet formed a periderm (image below still shows the endodermis in *ctr1-3*). Therefore, it is difficult to draw a conclusion from analysis using *ctr1* mutants and we think it is not meaningful to add their analysis in the manuscript. However, we have now increased the ethylene signalling in other way: we performed ethylene treatment (in addition to the existing ACC treatment data) and found a reduction in *PER15* and *PBP1* induction and suberized cell formation at the wound site (Extended Data Fig. 2l-o, Line 87-88, 92-93).

Left: Comparison of overall growth between wild-type and *ctr1* mutant. *ctr1* mutant shows severe growth defects. Images from Cheng et al 2009.

Right: Cross sections of 14-day-old wild-type and *ctr1-3* roots. While the periderm is established in wild-type, there is still the endodermis and no periderm formation in *ctr1-3* mutant. Suberin was stained with Fluorol yellow. Scale bars, 50 μm .

Referee #3 (Remarks to the Author):

This work from Lida et al. presents intriguing data sets on periderm regeneration, using older Arabidopsis root systems as a model. Based on their observations that treatment with the ethylene precursor ACC is able to suppress wound-induced periderm regeneration, the authors suggest an elegant model whereby wounding lowers ethylene levels in the tissue, through release of this gaseous hormone into the environment. This is suggested to serve as a signal for the initiation of periderm regeneration. As second signal required for regeneration is proposed to be the entry of oxygen into the hypoxic environment in the inner root tissue layers. This is a very elegant and intuitive model and the authors provide a set of well-executed experiments in support of this model, demonstrating the positive or negative effects of oxygen or hypoxia, respectively, on periderm regeneration, in combination with ethylene. The authors genetically manipulate hypoxia and ethylene signaling and provide measurements of ethylene release after wounding. They also provide data using reporter lines that can serve as proxies for oxygen status in the plant.

While I follow the authors in their assertion that they have identified a monitoring system for periderm integrity, I think they should acknowledge that the two signals – lower ethylene, increased oxygen – are clearly not sufficient to induce periderm formation/regeneration. If this were the case *ein2* ethylene signaling mutants should be “hair-triggered” to form periderm and increasing oxygen levels should lead to spontaneous/excessive periderm formation in roots, for example. This does not seem to be the case? In this context, I was surprised that the authors did not discuss their own previous paper on epidermal cell fate acquisition in any detail. In this previous paper in Nature Communications, the authors demonstrate that the acquisition of the epidermal cell fate specifier ATML1 by cortical cells is due to a release of pressure

upon ablation of the overlying epidermal cells. This is based on the observation that application of pressure is sufficient to suppress ATML1 expression in cortical cells at the wound site.

Did the authors investigate whether pressure release is an additional signal that is required for periderm regeneration? I think it would be logical to present some data on this in this story.

Authors: We appreciate this comment. We agree with the referee that the gases, ethylene and oxygen, are not the only regulators for regeneration. Further, we are grateful that the referee raises our previous work on epidermis regeneration. **SENTENCES REDACTED**

To provide sufficient evidence just for the mechanical stress aspect, comprehensive experiments are required, which necessitates a full manuscript to detail the methodology, results and analysis, just to show convincingly the role of the mechanical stress. Therefore, we would like to leave the mechanical stress aspect for future studies. However, as the referee suggested, we have now included new text in the Discussion to present the other possible pathways involving periderm regeneration (Line 279-282).

FIGURE REDACTED

I would also have loved to see experiments in high oxygen conditions (pure oxygen? Or air with increased oxygen levels?). This would allow to test whether oxygen increases can enhance periderm regeneration, or compensate for lower ethylene signaling, which would further strengthen their model. The authors present some data on aerated whether oxygenated 0.5 MS, but I don't find the effects terribly impressive and I wondered whether increased oxygen in a non-liquid environment would have stronger effects?

Authors: We tried higher oxygen level in air (50%) and found that it is unnaturally high level and thus causes pleiotropic defects in plant physiology and development due to oxidative stress (see seedling pictures below). However, we have new data to show that supplementing liquid media with air or oxygen promotes suberized cell formation, not only *PER15* induction, in *ein2* mutant under submergence (new Extended Data Fig 7g-j, Line 203-208), thus further supporting the role of oxygen in periderm regeneration.

Additional comments:

- I wonder if there is any way to get around the problem that wounding induces ethylene – because of this, it is difficult to claim that enhanced ethylene diffusion across the wound is occurring, since it could simply be due to the initially enhanced ethylene production? Have the author tried a pre- or co-treatment with ethylene biosynthesis inhibitors? Like this any increase in ethylene release after wounding would be more certain to be due to enhanced diffusion?

Authors: Thank you for this comment and suggestion for experiments. We detected ethylene signalling activity in the intact tissues (Extended Data Fig. 3a), and now provided new data that the signalling level in the intact tissue is reduced upon ethylene biosynthesis inhibitor treatment (Extended Data Fig. 3b). These results demonstrate that ethylene is accumulated in the intact secondary tissue. The emitted ethylene could originate from both accumulated and newly synthesized ethylene. To study the role of newly synthesized ethylene, we used AVG to inhibit ethylene biosynthesis. These treatments did not prevent barrier formation, in fact, the treatment led to enhanced suberization during regeneration (Extended Data Fig. 9c–h, Line 224–230). This further supports our idea that wounding-induced ethylene leakage, and the subsequent reduction of the signalling level, promotes barrier regeneration. It also supports the idea that ethylene biosynthesis and accumulation after the wounding functions as monitoring system for barrier integrity.

- The authors base a lot on *PER15* as a periderm marker and I wonder how specific this marker is. In Ext Data 4 they use *PBP1*, which also seems to work very well and I think it should be used as a second marker in some central experiments, such in Fig. 2G, 3D

Authors: In addition to the *PBP1* data in Extended data Fig. 4 (now Extended Data Fig. 2b), we have presented the *PBP1* reporter data also in Extended Data Fig. 5d, and newly made data in Extended Data Fig 2d,m, 9d,g. In general, *PER15* and *PBP1* behaved similarly, except after AVG treatment *PER15* was slightly but significantly reduced, while *PBP1* was not significantly reduced. Additionally, we have now conducted staining of suberized cells for additional critical experiments (Extended Data Fig. 7g–j). Together, these data show that *PER15* and *PBP1* expression, and suberin deposition correlate well in our experiments.

- Why are the results for *etr1* in Ext. Data 15 different from *ein2*? In ext data 5 *etr1* is as strong as *ein2*.

Authors: We are not sure why this is the case, however, we strongly suspect that the differences in response are attributed to the distinct nature of these two mutants and the varying sensitivities of the assays we employed. While *ein2* is a non-redundant loss-of-function mutant of ethylene signaling, *etr1-3* is a redundant, dominant negative mutant. The mutated *ETR1* receptor is active regardless of ethylene level, causing ethylene insensitivity. However, when *ETR1* expression is low in certain cell types, these cells would be able to respond to ethylene even in the presence of *etr1-3* mutation by the remaining ethylene receptors. Therefore, it is logical that the *ein2* mutant shows stronger ethylene insensitivity compared to *etr1-3* mutant. We explain this in Figure legend “The different response between *ein2-1* and *etr1-3* is likely due to the distinct nature of the two mutants. Because *etr1-3* is a dominant negative mutation, the negative effect of *etr1-3* in ethylene signalling is lower in cells where the expression of other ethylene receptors is dominating.”. Nevertheless, the *etr1-3* mutant also showed a similar trend to *ein2* (Extended Data Fig. 7a–f, 9a,b), and we now also provide new data showing a significant increase in suberized cell formation in the

etr1-3 mutant under the aerated or oxygenated liquid medium conditions, in contrast to the wild-type (Extended Data Fig. 7g–j, Line 203–208).

- I think Ext Data 6 would benefit from additional quantification of the signals. It is hard to see how the authors simply classified the quantitative difference in signal strength as different categories.

Authors: We agree with this comment. The quantification data for Extended Data Fig. 6 was previously shown together with time course data in the Extended Data Fig. 16e. For the convenience of readers, we have added quantification data in the new Extended Data Fig. 3e.

Referee #4 (Remarks to the Author):

The manuscript by Lida et al. proposes that plants can monitor the integrity and regulate the repair of barrier tissue via gas diffusion. Data is presented to support three main conclusions: that periderm integrity is sensed by ethylene leaking out and oxygen entering (following wounding); that this is mediated by attenuated ethylene signaling and hypoxia signaling and this in turn mediates periderm regeneration.

The novelty here lies in linking both ethylene and oxygen dynamics to barrier formation following wounding (although the data presented is not completely convincing for the claims made— see my comments below).

Also, there is previous work that has established the link between ethylene, wounding and periderm/suberization formation in sweet potato and potato (which I don't believe the authors cited).

Reference: <https://doi.org/10.21273/HORTSCI.24.5.805> (St Amand and Randle, 1989);

<https://doi.org/10.1016/j.postharvbio.2004.05.012> (Lulai and Suttle 2004)

Authors: As the referee notes (and as also proposed in one of these two referenced papers), the prevailing concept is that wound-induced ethylene production plays a key role in PROMOTING wound healing by facilitating the regeneration of periderm. In this manuscript, we show the opposite: ethylene INHIBITS periderm formation, at least in the secondary tissues of *Arabidopsis* root. We are confident that both our previous and new data included in this revision provide strong support for this conclusion. Our new data, for instance, demonstrate that treatment with an ethylene biosynthesis inhibitor does not prevent regeneration; instead, it actually promotes regeneration, providing further support for our conclusion (New Extended Data Fig. 9c,d). We cannot rule out the possibility that ethylene biosynthesis is also induced from its basal level after wounding in the *Arabidopsis* root (and such induction is likely). However, it is clear that ethylene does not promote regeneration. Instead, ethylene accumulation serves as an indicator of barrier integrity restoration, ultimately leading to the cessation of suberization. To the best of our knowledge, this is the first report demonstrating genetically that ethylene diffusion is utilized to monitor and re-establish barrier integrity. This paper may even be the first one to hypothesize such a mechanism, although it is possible that someone has proposed the idea during the decades-long history of studying ethylene and wound healing.

The previous studies in sweet potato and potato mentioned by the referee used various ethylene inhibitors and suggested ethylene either promotes or plays no role in periderm barrier formation upon wounding. Thus, the general relationship between ethylene and periderm regeneration is unclear. Since these studies are based on non-genetic models, the genetic evidence is lacking. The defects in periderm formation observed with the inhibitor treatment may result from overall growth inhibition caused by these chemicals, if not properly controlled. Additionally, the periderm regeneration mechanism likely differs between the species and organs. Now we have elaborated the previous findings and our findings in the beginning of Discussion (Line 266–272).

Comments:

1) Periderm re-establishment: the authors use periderm reporter genes, phellogen reporters and suberin and lignin to demonstrate periderm recovery following injury. Suberin and lignin is very clearly visible, but the markers are harder to see and I see no quantitation for these reporter signals. This is needed.

Authors: The referee is right – some of our figure panels needed some improvements. We have now adjusted the signal of the reporters in Fig. 1c, Extended Data Fig. 1a. Additionally, we have now quantified the fluorescence levels in Extended Data Fig. 1c. Finally, all the new data provided for this revision have brighter reporter signal and quantification.

2) To link ethylene function to periderm regeneration, the authors assess the induction of PER15 as a periderm marker (to find that ACC suppresses expression); Here the authors use the wounded 'mock' as controls. Since wounding itself triggers ethylene production, a better control to see 'ethylene/ACC' triggers up/downregulation of marker genes is to also treat an unwounded root (with/without the hormone treatment). This goes also for any readout (eg suberin) where ACC is added to injured roots. The response observed in these roots is likely that to a very high concentration of ethylene (wounding induced ethylene + ACC). And it is well known that ethylene like most hormones elicits dose dependent responses.

Authors: Our ethylene reporter line showed that ethylene signalling level is reduced upon wounding. Therefore, wounding does not lead to the upregulation of ethylene signalling level (triggered by ethylene production) as the referee reasoned. ACC treatment with unwounded roots is a good experimental setup to test whether ethylene signalling is involved in both normal periderm formation and periderm regeneration. We originally showed that ACC treatment did not affect normal periderm formation based on suberized cells in Extended Data Fig. 14 (we have also revised this data according to the comment from the referee 2 (new Extended Data Fig. 2g)), indicating that there are other signals during endogenous development. Now, we have conducted ACC treatments both for unwounded *PER15* and *PBP1* reporter lines. Consistent with the effect on suberization, *PER15* and *PBP1* expression were not affected by ACC treatment in unwounded roots (Extended Data Fig. 2c,d,f, Line 83-85). We have also performed ACC treatment with unwounded ethylene signaling reporter and showed that ACC treatment strongly increased ethylene signalling level (Extended Data Fig. 3b,f,g, Line 106-109). For the dose dependent responses, please refer to our answer to the comment 9.

3) Since PER15 is primarily used throughout the manuscript as a readout for periderm regeneration, can the authors show that per15 mutants are incapable of periderm regeneration?

Authors: In this manuscript, we are not investigating the role of peroxidases in periderm regeneration; rather, we are using *PER15* as a marker to monitor the early steps in periderm regeneration. Initially, we analyzed several periderm markers to identify one that is rapidly and consistently induced by wounding. From this analysis, *PER15* emerged as the most suitable marker. However, the referee is correct in noting that the induction or reduction of *PER15* does not necessarily indicate that periderm regeneration will occur or fail to occur, respectively. This is the case even if *per15* ko mutant had a defect in regeneration. Therefore, we have now analyzed suberization in the submergence experiments where we found recovery of *PER15* induction in *ein2* background when oxygen is supplemented to liquid medium. Consistent with *PER15* recovery, defects in suberization in *ein2* mutants grown under liquid medium without gas supplementation were rescued when submerged in aerated or oxygenated liquid medium (Extended Data Fig. 7h,j, Line 203-208). We have also examined key experiments with another periderm reporter, *PBP1*

(Extended Data Fig. 2b,d,m, 5d, 9d,g). In summary, we found that *PER15* expression, *PBP1* expression and suberization correlate well.

- If ethylene is a negative regulator of periderm formation ethylene addition should prevent normal periderm regeneration. This is tested by addition of ACC and the finding that the number of suberized cells is lower or discontinuous. In Figure 2 a, b, I missed the data for controls (unwounded roots). Also here an important experiment is to not only add the signal (in this case ACC) to wounded roots but remove the signal (ethylene inhibitor). Adding ACC, to wounded roots that already are producing ethylene would lead to very high levels of ethylene and might confound the output (hormones can have opposite effects depending on concentration). With these experiments, one would expect to see opposite trends for periderm markers and importantly suberisation. The cleanest is a chemical approach. Ethylene mutants tend to confound effects due to their strong developmental effects (unless these are conditional mutants). Also linked to this comment, considering that one of the main conclusions of this work is the involvement of ethylene in barrier repair, I find it very puzzling that ethylene mutants have no suberin phenotype (Extended data fig 17). This suggests that while ethylene(ACC) influences the regulation of *PER15*, ethylene signaling is not required for barrier regeneration. Even if hypoxia signalling is active in these mutants, one would expect some contribution of ethylene signaling Did the authors try block hypoxia signalling? For example, by providing hyperoxia? What would happen if lanolin is applied to these mutants?

Authors: Thank you for the comments.

As suggested by the referee, we took a chemical approach to reduce ethylene signalling levels using AVG (biosynthesis inhibitor) and silver nitrate (Ag; signalling inhibitor). Induction of *PER15* and *PBP1* was not significantly affected by these treatments, except that *PER15* induction was slightly reduced upon AVG treatment (Extended Data Fig. 9c,d,f,g), suggesting that further reduction in ethylene signaling levels does not enhance the induction of these genes. However, we found that suberized cell formation is enhanced upon AVG or Ag treatment (Extended Data Fig. 9e,h). Furthermore, we now found an increased suberized cell formation in *ein2* mutant 6 days after the injury (Extended Data Fig. 9a,b). Previously, we did not observe significant difference 4 days after injury when the fully sealed barrier is just completed (Extended Data Fig. 9b). Thus, this finding supports the idea that ethylene level increase after sealing is required to precisely terminate the suberization process. These new data are described in Line 223-224. These results further support our model that once ethylene levels return to pre-injury levels, regeneration is halted. Chemical inhibition of ethylene biosynthesis or signaling, or genetic disruption of ethylene signaling, prevents the achievement of pre-injury signaling levels. As a result, regeneration continues even after barrier re-establishment, leading to the overproduction of cork cells.

As described in the comment 2 above, we did not find defects in normal periderm formation even when ethylene and hypoxia signalling are high, indicating that there are other signals for normal periderm formation (Extended Data Fig. 2g). In fact, referee 1 suggested the appropriate term to describe the roles of the two gases: low ethylene and hypoxia levels create permissive environment for regeneration. This may explain why high ethylene and hypoxia levels hinder regeneration (non-permissive conditions), while low ethylene and hypoxia levels (permissive conditions) enable the periderm regeneration machinery to form the barrier without overly excessive buildup. We have now elaborated this aspect in the Discussion (Line 282-283).

4) The functionality of the barrier tested for ACC and as shown in extended data figure 2c is not very convincing i.e. ACC treatment does not really reverse the 4 dai trend. Should the ACC treatment not be done earlier (unless I read the methods wrong and it is only the 4dai measurements that are shown).

Authors: Since the effect of ACC on periderm regeneration is not fully penetrant, proPXY:GUS signals increase only mildly in ACC-treated roots. Also, since the promoter activity of PXY is decreased by ACC treatment (Extended Data Fig. 1d,e), this analysis underestimates the effect of ACC treatment. Nevertheless, we found an increase in GUS signal intensity upon ACC treatment, indicating that this assay detected the defects in barrier functionality in ACC-treated roots. ACC treatment was initiated right after the wounding (Methods, Line 344-345).

5) In Figure 2f the EBS:GUS marker is used as a readout of ethylene signalling (and in the extended data RPS5A line). Because there seems to be a discrepancy between ethylene emitted and ethylene signaling, there should be similar time points for the two. For example ethylene emission steadily increases over time (as measured upto 1dai), but ethylene signaling as indicated by the two ethylene reporter lines suggests downregulation. In Figure 2f 0dai shows intense staining -indicating a very rapid ethylene signaling response. An important missing control here is the unwounded root, that should theoretically show no staining. Also missing are the earlier time points that were used for measuring ethylene emission.

Authors: Thank you for this comment, which revealed that we failed to describe properly a couple of issues, and this has caused some misunderstanding. "a discrepancy between ethylene emitted and ethylene signaling" is in part because of different time scales between these two monitoring methods. While GC can detect immediate changes in ethylene levels, genetic reporters like *EBSx5:GUS* and *RPS5A:erVenus-EBF1UTR* require longer times to react changes in ethylene concentrations. In Fig. 2f, we did the staining immediately after the injury. We need to do injury for all the GUS experiment, because intact periderm severely delays GUS substrate intake, causing large variation in the results. We do not think *EBSx5:GUS* expression at 0 dai is from very rapid ethylene signaling response as the referee mentioned, but the signals reflect the steady state level of ethylene signaling in intact tissue. We also used *RPS5A:erVenus-EBF1UTR* in which we don't need to wound to detect the signal. Consistently, high ethylene signaling levels are also detected in intact roots of *RPS5A:erVenus-EBF1UTR* reporter line especially in distal phloem region adjacent to wound, where regeneration occurs. Please, remember that *RPS5A:erVenus-EBF1UTR* is a reverse reporter: low level of fluorescence indicates high ethylene signaling levels (see Extended Data Fig. 3). We have now further highlighted the 'reverse nature' of the reporter in Line 105-106. To detect the reduction in *EBSx5:GUS* signals, it takes some further delay since the existing GUS enzyme is stable even when the transcription of new GUS has already stopped. In this aspect, *RPS5A:erVenus-EBF1UTR* is more suitable because the reduction in ethylene signaling is detected as the increase of Venus signals. Yet, we agree with the referee that what we explain above does not explain entirely the large time difference between the emitted ethylene (3h) and the decrease in ethylene signaling (24h). Thus, we have now analyzed *RPS5A:erVenus-EBF1UTR* at earlier time points after injury, and we found that erVenus-EBF1 signal increases (i.e. ethylene signaling level decreases) already 8h after the injury (Extended Data Fig. 4b). Therefore, considering also the above-mentioned differences in detection times between these two methods (GC and genetic reporter), we believe our data support well the idea of simultaneous emission of ethylene and reduction of ethylene signaling inside of the root. We have now elaborated these aspects better in Results (Line 120-124). Because of the above-mentioned shortcomings of *EBSx5:GUS* reporter (also pointed out by the referee in comment 10), we decided to exclude *EBSx5:GUS* data from this manuscript.

6) I have a problem with the term 'ethylene leaks out'. This suggests a containment of the hormone under normal conditions and a breach (wounding) resulting in the hormone leaking out along a strong concentration gradient. This should then occur instantaneously within very short durations, unlike the decline in ethylene signaling seen here (also why it is important to have ethylene signaling monitored dynamically). This is assuming that under control conditions there are high levels of the hormone compared to wounded. This is not the case (as also the data here points out; or I would recommend additional measurements eg: ethylene marker genes). Unless the authors interpret the increased ethylene emission as ethylene 'leaking' out. It is well established that wounding upregulates ethylene biosynthesis (and therefore ethylene emission), which is what I interpret the data here as. The authors should check this (and untangle ethylene biosynthesis and signaling) using ethylene biosynthesis mutants.

Authors: First of all, our initial data show that there is a higher ethylene signalling level in the secondary tissue in intact roots (Extended Data Fig. 3a,e; comparison between upper (control without EBF1-3'UTR) and lower panel). Please note that *RPS5A-erVenus-EBF1UTR* is "reversed" reporter: in high ethylene levels, Venus signal is low (Extended Data Fig. 3a-c,e-i, root apical meristem and secondary tissue). We have now further highlighted the reverse nature of the reporter in Line 105-106 ("Thus, low Venus fluorescence indicates high ethylene signalling levels."). Second, we have now carried out time course observation of the ethylene reporter line and showed that ethylene signalling level is reduced already 8 hours after the injury (i.e. Venus signal is enhanced at the wound site), supporting that, following injury, ethylene emission causes rapid reduction in its signalling level in the tissue (Extended Data Fig. 4b, Line 120-124). We have already highlighted that *ein2* and *etr1-3* mutants do not prevent regeneration, in fact, we provide new data that *ein2* showed enhanced barrier formation (Extended Data Fig. 9a,b, Line 223-224). Furthermore, now we have new data to untangle ethylene biosynthesis and signalling using chemical inhibitors. To exclude the role of newly synthesized ethylene, we used AVG or Ag to inhibit ethylene biosynthesis or signalling, respectively. These treatments did not prevent barrier formation, in fact, the treatment led to enhanced suberization during regeneration similar to the phenotype in *ein2-1* (Extended Data Fig. 9a,b,e,h, Line 230-231). This further supports our idea that wound-induced leaking of ethylene, and the subsequent reduction of the signalling, promotes barrier regeneration.

For ethylene emission and ethylene marker changes, please refer to our response to the comment 5.

7) I'm a bit confused between the correlation (or lack of) of ethylene emission and ethylene signaling. Can the authors (repeating a comment I made above) demonstrate using similar time points for ethylene emission and ethylene response markers/reporters? I would also like to see ethylene signalling output for the lanolin experiments. Ethylene can feedback inhibit itself, which means high ethylene levels can downregulate ethylene responses and this would reveal that.

Authors: For ethylene reporter expression at earlier time points, please refer to our response to the comment 5 and 6. Also, we have shown that Vaseline treatment maintained high ethylene signalling level upon injury (please refer to Extended Data Fig. 5a,b). This is inconsistent with the idea that high ethylene levels inhibit ethylene responses. In addition, we treated ethylene reporter lines with low and high ACC concentration and found that the upregulation of ethylene signalling level correlated well with the ACC concentration (please refer to our response to the comment 9). Therefore, the results indicate that blocking ethylene diffusion maintains both the ethylene level and its signaling level at the same level as in the intact root.

8) Although the authors have taken care to show that ACC effects are not independent of its conversion to ethylene, I would suggest an experiment with ethylene gassing (like what they have done for hypoxia) and observe marker gene and suberisation.

Authors: We performed ethylene treatment, and this resulted in similar responses as ACC treatment (Extended Data Fig. 2l-o, Line 87-88, 92-93).

9) To resolve the dose dependent effects of ethylene, the authors could do an experiment with a low and high concentration of ethylene (on wounded roots) and then assess output.

Authors: We have now treated the ethylene reporter line with low (2 μ M) and high (10 μ M) ACC concentrations upon wounding and found that ethylene signalling level exhibited an increase in accordance with the ACC concentration. Therefore, the ACC treatment of wounded roots does not trigger a negative feedback effect to down-regulate ethylene signaling output, as the referee expressed concern about in comment 2 and 7 (Extended Data Fig. 4a and added text in its legend, “*RPS5A:erVenus-EBF1UTR* signals showed a gradual decline as the ACC concentration increased, confirming that ACC treatment increases ethylene signalling level in wounded roots.”)

10) For both ethylene reporters (especially since one is newly synthesized reporter line and the other famously leaky), a minimum positive control is needed (plants treated with and without ethylene/with and without ethylene inhibitor).

Authors: As the referee pointed out, *EBSx5:GUS* signals were variable in roots, and like explained above, cutting the root just before the analysis is needed to get GUS substrate in. For these reasons, we decided to exclude all the *EBSx5:GUS* from this manuscript. Since *EBSx5:GUS* is not reliable, we have focused on the newly generated reporter line, *RPS5A:erVenus-EBF1UTR*. We have now provided a detailed validation of our new ethylene reporter line, both in the root meristem (since there, ethylene signaling has been studied in detail before) and in the mature part of the root. Our results showed that *erVenus-EBF1UTR* reacts to ACC and ethylene inhibitor treatment, as previously reported for ethylene signaling reporters in the root meristem (Li et al., 2015, Cell (DOI: 10.1016/j.cell.2015.09.037)). Additionally, it behaves consistently in the mature part of the root, as explained above. These data are presented in new Extended Data Fig. 3b,c,f-i (Line 106-109).

11) The authors only look at O₂ as an additional gas, and I guess the choice was due to its recent emergence as an important developmental regulator. However there are other gases that could be physiologically relevant eg: NO, CO₂. Did they not test those, or were the results negative – could the authors comment?

Authors: This is indeed an interesting idea. In addition to ethylene, we tested ABA and JA (MeJA, volatile organic compound) with no effect (Extended Data Fig. 1g,h). We have not tested NO and CO₂, since we observed complete inhibition of root periderm regeneration under high ethylene and hypoxia. However, it is intriguing to explore the other gases in shoot barrier regeneration since the barrier re-establishment is likely mediated by other gases than ethylene or oxygen (Extended Data Fig. 10). We consider this exploration for future studies.

12) Throughout the study the plants use vary in age between 14-21 old. Is there a reason for this choice? Plants can vary in their response depending on age.

Authors: For examining normal periderm, we used 14- or 16-day-old seedlings since periderm is established at that age. For assessing regeneration, we usually used 17- to 21-day-old seedlings. The comparison was performed among the samples whose age was the same at the time of analysis (e.g., 21+0d vs 19+2d vs 17+4d). Considering secondary tissues at distinct ages show quite different morphology, this approach can compare samples with less morphological differences. We have now elaborated this in the Methods (Line 309-311).

13) Figure 3b – did the authors look at shorter time points for the PCO response? Again in keeping with gas diffusion, I would expect the effects to be very quick, i.e. within a few hours. Figure 3c, could the authors recalibrate the scale to pO₂ in %? This would help non-experts relate the concentration changes easier. Was the choice of 5% Oxygen in the hypoxia experiments guided by the oxygen levels observed in 3c?

Authors: As we explained in our response to the comment 5, genetic reporter lines take time to respond to the changes, especially when the signal is expected to reduce (reduction depends on the half-life of the YFP, and Venus-YFP has been engineered to be very stable). Anyway, our microsensor showed a rapid increase in oxygen levels, very soon after removal of periderm (Fig. 3c). We selected 5% oxygen because, under these conditions, plants can sense hypoxic stress but are still able to grow. This is important because when the concentration is too low, growth and physiological processes will be impaired regardless of its relevance to hypoxia signaling. We have now converted the oxygen concentration into % in Fig. 3c.

14) The data showing that reduced hypoxia signaling is required for periderm generation is not very convincing. Difference in PCO1 signal intensity is marginal. Two mutants are used that have constitutive hypoxia signaling. While *ate1ate2* show the expected results, *prt6* response is similar to wild-type. Fig 3: Why not include the imaging of *prt6-5*? And how can the difference between *ate1;ate2* and *prt6-5* be explained? Since these are in the same pathway and logically should have the same phenotype?

Authors: The difference of PCO1 signal intensities between 0 and 2 dai is significant, and we have repeated this experiment three times always with significant differences. We agree that the difference is not massive, but the whole point of the paper is that while elevated hypoxia or ethylene signaling alone causes weak but significant regeneration defects, their combination leads to nearly complete loss of periderm regeneration (Fig. 3f). We have previously reported that *ate1;ate2* mutant shows more enhanced hypoxia response compared to *prt6* (Giuntoli et al., 2017, Plant Cell and Environment (<https://doi.org/10.1111/pce.13037>)). Also, since a recent report identified *BIG* gene as a redundant gene for *PRT6* (Zhang et al., 2024, Plant Cell (<https://doi.org/10.1093/plcell/koae117>)), the phenotype difference is likely explained by *PRT6*'s redundant partner. Furthermore, we have performed new experiment and showed supplementing oxygen in liquid medium rescued cork cell formation in ethylene insensitive mutant, further demonstrating the role of oxygen in periderm regeneration (new Extended Data Fig. 7, Line 203-208). Since *prt6* data especially with ACC treatment is just redundant with *ate1;ate2* data and there is a limited space in the main Figure 3, we are keeping the *prt6* data in the Extended Data Fig. 6e.

15) Throughout the manuscript the authors generalize the observations to attribute it to 'gas diffusion'. This should be altered to specify oxygen and ethylene, since there are other gases that were not tested here.

Authors: Relevant point. We have now modified the text accordingly where we need to specify the diffusing gases are ethylene and oxygen (Line 213, 219, and 236).

16) A general thought about the relevance of this mechanism in natural environments: if I understand correctly the authors pull the seedlings out of its substrate to make a cut which is then exposed to the atmosphere. In natural conditions in a soil environment, there would not be such a free exchange of gases. How relevant would 'gas diffusion' be for a plant to regulate barrier properties?

Authors: It has been demonstrated that primary root meristems sense soil compaction by detecting the diffusion of ethylene in the soil (Pandey et al., 2021, Science (DOI: 10.1126/science.abf3013)). While ethylene emitted from root tips diffuses well through normal soil, ethylene diffusion is blocked in compacted soil, leading to an increase in ethylene level in root tips and an attenuation of root growth. Therefore, there are spaces for ethylene to diffuse in normal (i.e. not compacted) soil environment. The same paper also demonstrates that there is no hypoxic condition, even in compacted soil. Additionally, extreme hypoxia (anoxia), such as that caused by flooding, can lead to the death of root systems. Thus, to survive, plants need to obtain oxygen from the surroundings soil and surrounding soil must have oxygen available for roots, through root tips. Root meristems don't contain barrier tissue: endodermis is formed in the differentiation zone, and periderm is replacing endodermis as barrier tissue later on. Thus, while root tips (meristem region) can freely exchange gases, more mature parts of roots limit gas exchange. And the latter is one of the key findings of our paper

17) The last statement line 237, that "monitoring of barrier integrity via gas diffusion" is also relevant in stems is fine, but the data actually does not reveal what these gases are. Data presented does not support a strong role for either ethylene or oxygen. The lack of PCO induction is perhaps not surprising due to the photosynthetic nature of shoot tissue; Again an important time point for 4g are the unwounded tissues.

Authors: Application of Vaseline or lanolin to the shoot wounds supports the idea of gas diffusion also in shoots. There are several intriguing candidate volatile molecules such as CO₂, NO, JA, and SA. Also, we are currently working on water vapor as a candidate. Because there are many candidates, we will leave the identification of gases for shoot barrier regeneration for future studies. We now replaced *EBSx5:GUS* data with our new ethylene reporter data in the stems and provided its expression in the intact tissues (Fig. 4g, Extended Data Fig. 10e,h,i).

Dear Referees and Dear Editor,

We appreciate the referees' comments. Please find below our response.

Referee #1 (Remarks to the Author):

The authors have addressed all my concerns with either new experiments or reasonable explanations addressing my concerns. I think this is a valuable manuscript and novel finding in the field of plant regeneration in terms of uncovering a new primary signal that tells the plant it is injured in order to invoke repair.

Referee #2 (Remarks to the Author):

This MS has been revised and greatly improved based on my previous comments and I have no further comments.

Referee #4 (Remarks to the Author):

I would like to thank the authors for clarifying my concerns and strengthening their conclusions based on additional experimental evidence.

I do not have any further comments!

Authors: Thank you for the encouraging words.

Referee #3 (Remarks to the Author):

The authors have addressed all my comments/concerns in this carefully revised version of the manuscript. I have no further comments.

Just one remark for the authors' consideration: If higher oxygen levels cause pleiotropic effects on plant growth, maybe this is simply due to enhanced photo-respiration and could be alleviated by simultaneously raising CO₂ concentration, such that the ratio of CO₂/O₂ remains constant?

Authors: Thank you. The reviewer's point is valid — the observed toxicity could indeed have been a result of photorespiration. Alternatively, toxicity of 40% O₂ could be linked to ROS production/oxidative stress, independent of photorespiration. We are technically limited in conducting the proposed experiments, as our current setup only allows mixing two gases (air or pure oxygen with nitrogen). Setting up a system with elevated O₂ and CO₂ and testing them with different concentrations would take several months minimum, so we would not like to delay the publication any further. We believe that we have already addressed the role of O₂ levels in periderm regeneration sufficiently.

Authors: We have also generated new data. To test how quickly hypoxia response is attenuated after the injury, we examined the activities of *PCO1* and *PCO2* promoter within 24 h after the injury. Our new data shows reduced *PCO1* and *PCO2* expression already at 16 h after wounding, further supporting a quick reduction in hypoxia response after injury. We believe this new data would benefit the manuscript and we included it in Extended Data Fig. 6c.

Additionally, during the revision, we noticed that a statistical test description in Extended Data Fig. 9b was not precise. We performed pairwise comparison, not multiple comparisons. We revised the figure legend accordingly. The difference between the wild-type and *ein2* is statistically significant.

The final version of the rebuttal letter has been reviewed by the referees.